# Diffusion Models Are Statistically Optimal for Learning Low-Dimensional Multi-Modal Distributions

**Jingda Wu** [1]   **Changxiao Cai** [1]

## Abstract

Score-based diffusion models have demonstrated remarkable empirical success in learning high-dimensional distributions, particularly those exhibiting low-dimensional and multi-modal structures. However, theoretical understanding of their statistical efficiency remains limited. Existing theories typically rely on strong regularity assumptions, such as uniformly bounded densities or globally smooth score functions, which fail to capture such intrinsic structures. In this work, we study the sample complexity of diffusion models for learning distributions supported on a union of low-dimensional subspaces. Assuming that the data distribution within each subspace is subgaussian, we show that diffusion models require at most the order of $\widetilde{O}(\varepsilon^{-k \vee 2})$ (up to some logarithmic factor) samples to achieve $\varepsilon$ sampling error in 1-Wasserstein distance, where $k$ is the intrinsic dimension. This near-optimal convergence rate depends only on the intrinsic dimension and significantly improves upon prior theoretical guarantees that suffer from the curse of dimensionality. Notably, our analysis applies to a broad collection of distributions without imposing smoothness, bounded-density, or log-concavity assumptions. Overall, our results show that diffusion models can statistically adapt to intrinsic low-dimensional structure while naturally accommodating multi-modal data, offering a rigorous theoretical justification for their success in complex high-dimensional learning tasks.

[1]Department of Industrial and Operations Engineering, University of Michigan, Ann Arbor, USA. Correspondence to: Changxiao Cai <cxcai@umich.edu>.

*Proceedings of the 43$^{rd}$ International Conference on Machine Learning*, Seoul, South Korea. PMLR 306, 2026. Copyright 2026 by the author(s).

## 1. Introduction

Score-based diffusion models (Sohl-Dickstein et al., 2015; Song & Ermon, 2019) have achieved state-of-the-art performance across a wide range of generative modeling applications, including image and video generation (Ho et al., 2020; Ho et al., 2022), signal processing (Song et al., 2021), and language modeling (Austin et al., 2021; Nie et al., 2025). At a high level, diffusion models generate samples by starting from Gaussian noise and iteratively denoising via a learned reverse-time diffusion dynamics. This procedure relies on accurate estimation of the *score function* (the gradient of the log-density) along a forward noising process.

Diffusion models can be viewed as unsupervised distribution learners and samplers—given finite training samples from an unknown data distribution, they aim to learn the underlying law and generate new samples that faithfully follow it. This perspective raises a fundamental statistical question: *how many training samples are needed for diffusion models to accurately learn the underlying data distribution, and can this sample complexity match the information-theoretic limit?* At a conceptual level, diffusion-based sampling is inherently two-stage: it first uses training data to estimate the time-indexed scores along a forward diffusion process, and then plugs these learned scores into an iterative sampling procedure to generate an output. Therefore, addressing the above question calls for sample complexity guarantees that jointly control both the score estimation error and the error accumulated during sampling.

**Leveraging intrinsic structures.** Motivated by this, a growing body of statistical theory has been developed to understand the sample complexity of diffusion models (Shah et al., 2023; Oko et al., 2023; Chen et al., 2024; Cole & Lu, 2024; Li et al., 2024; Dou et al., 2024).

For a broad class of $d$-dimensional distributions with $\beta$-Hölder smooth densities (without assuming smooth scores or log-concave/uniformly-bounded densities), state-of-the-art theory shows that both DDPM (Zhang et al., 2024) and DDIM (Cai & Li, 2025) require on the order of (up to logarithmic factors)

$$\varepsilon^{-\frac{d+2\beta}{\beta}} \tag{1}$$

training samples to generate an output within $\varepsilon$ total variation (TV) distance to the target distribution. While this sample complexity is (nearly-)minimax optimal for general smooth-density classes, it suffers from the curse of dimensionality as the ambient dimension $d$ grows. Consequently, such guarantees fail to fully explain the empirical effectiveness of diffusion models in modern high-dimensional applications, suggesting that the worst-case bounds in (1) may be overly pessimistic for structured distributions arising in practice.

To narrow this gap, recent work has explored whether, and in what sense, diffusion models can exploit intrinsic structures underlying the data distributions. However, statistical theory of diffusion models for structured data distributions remains far from complete. Existing results in this direction typically focus on distributions supported on a *single* low-dimensional structure, such as a linear subspace or manifold (Chen et al., 2023; Tang & Yang, 2024; Azangulov et al., 2024; Yakovlev & Puchkin, 2025), a factor model (Chen et al., 2025a), or certain dependence structures (Fan et al., 2025). Although these works establish improved sample complexities governed by intrinsic rather than ambient dimension, they require strong assumptions on the data distribution. A prominent example is the requirement for the density to be uniformly bounded away from zero on its support. While this condition is standard in the nonparametric statistics literature and technically convenient, it excludes important multi-modal structures with well-separated components, where the density necessarily becomes small, or even vanishes, between modes.

More fundamentally, the prevailing "single manifold/subspace" paradigm limits our theoretical understanding of diffusion models' capabilities to learn heterogeneous distributions whose different modes concentrate near different low-dimensional structures. Such geometry is common in modern high-dimensional data, where distinct classes or clusters may occupy separate subspaces or manifolds (Vidal, 2011; Brown et al., 2022). As a result, existing theory still falls short of explaining the empirical effectiveness of diffusion models when learning low-dimensional, multi-modal distributions.

### 1.1. Main contributions

In this paper, we develop a statistical theory for diffusion models, aimed at understanding how many samples are required to learn low-dimensional, multi-modal distributions.

Concretely, we consider a target data distribution $p^\star$ supported on a union of subspaces (UoS), i.e.,

$$\text{supp}(p^\star) \subseteq \cup_{i=1}^{M} V_i,$$

where each $V_i$ is a linear subspace with dimension $k_i$. We denote by $k := \max_{i \in [M]} k_i$ the maximum intrinsic dimen-

sion. In addition, we assume that the restriction of the target distribution to each subspace is $\sigma$-subgaussian.

Under these two assumptions, we construct a kernel-based regularized score estimator $\widehat{s}_t$ for the score function of the Gaussian-smoothed distribution $p_t := p^\star * \mathcal{N}(0, tI_d)$ for any $t > 0$. Given $n$ samples drawn from the target distribution $p^\star$, we establish a finite-sample $L^2$ score estimation error bound:

$$\mathbb{E}\left[\|\widehat{s}_t(X) - s_t^\star(X)\|_2^2\right] = \widetilde{O}\left(\frac{1}{n}\left(\frac{1}{t} + \frac{1}{t^{(k\vee 2)/2+1}}\right)\right).$$

Here the expectation is taken over both the training data and $X \sim p_t$, where we write $a \vee b := \max\{a, b\}$.

Building on this score estimation guarantee, we prove that diffusion samplers require at most the order of (up to logarithmic factors)

$$\varepsilon^{-(k\vee 2)}$$

training samples to generate a sample that is $\varepsilon$-close in 1-Wasserstein distance to the target distribution $p^\star$. Importantly, this convergence rate depends only on the intrinsic dimension $k$, rather than the ambient dimension $d$, and matches the minimax optimal rate for learning a $k$-dimensional distribution (Chewi et al., 2024). Moreover, our theory requires only subgaussian tails on each subspace, without imposing any restrictive assumptions on scores or densities. As a result, our framework naturally accommodates multi-modal distributions with well-separated components.

Finally, we emphasize that the kernel-based score estimator developed in this paper is primarily a theoretical proof device, rather than a practical alternative to neural network (NN)-based score estimation. Nevertheless, our results provide an important step toward statistical guarantees for NN score-based diffusion models. In particular, they establish the achievability of the fundamental statistical limit and identify the structural properties that analysis of NN score estimators should capture, while also providing an explicit low-dimensional target for NN approximation. More discussion on extensions to NN-based scores can be found in Section 6.

### 1.2. Related works

**Statistical theory for diffusion models.** Recent work has begun to provide finite-sample guarantees for diffusion-based sampling by studying statistical bounds for score estimation error. Under strong density regularity assumptions (e.g., boundedness on compact domains), Oko et al. (2023) showed that neural-network-based ERM score estimators lead to minimax-optimal rates in both TV and $W_1$ distances when used with reverse SDE samplers. Using nonparametric

constructions under density lower-bound conditions, Dou et al. (2024) derived minimax-optimal score estimation rates and corresponding sampling guarantees. More recently, for subgaussian targets with $\beta$-Hölder smooth densities, Zhang et al. (2024) established minimax-optimality for DDPM using truncated kernel score estimators, and Cai & Li (2025) obtained an end-to-end minimax-optimal convergence analysis for ODE-based diffusion (DDIM/probability flow) by combining smoothed score estimation with convergence analysis of the sampling dynamics.

**Adaptation to low-dimensional structures.** For distributions supported on a linear subspace, Chen et al. (2023) established convergence rates governed by the subspace dimension under smooth score assumptions. For manifold-supported targets, Azangulov et al. (2024) proved analogous intrinsic-dimension rates, but the analysis requires controlling geometric approximation error (e.g., via Hausdorff distance) and typically relies on density lower-bound conditions on the support. Beyond geometric support constraints, Fan et al. (2025) obtained minimax-optimal rates for diffusion learning under structured dependence (exponential-interaction) models. In addition, Wang et al. (2024) analyzed mixtures of low-rank Gaussians, focusing on the special case of orthogonal subspaces. Boffi et al. (2025) showed that shallow NN-based diffusion models can provably adapt to hidden low-dimensional subspace structure under independent component data models and smoothness assumptions on the latent scores.

Complementary to the statistical perspective, a parallel line of work studies whether the sampling stage of diffusion models can automatically exploit low-dimensional data structure (Li & Yan, 2024; Liang et al., 2025; Potaptchik et al., 2024; Huang et al., 2024). These works show that the iteration complexity, the number of sampling iterations required to achieve a desired accuracy, also depends only on the intrinsic dimension rather than the ambient dimension. In addition, low-dimensional adaptation has also been investigated for discrete diffusion models when learning discrete distributions (Li & Cai, 2025; Zhao & Cai, 2026; Cai & Li, 2026; Chen et al., 2025b; Dmitriev et al., 2026).

### 1.3. Notation

For $a, b \in \mathbb{R}$, we denote $a \vee b := \max\{a, b\}$ and $a \wedge b := \min\{a, b\}$. For positive integer $M$, let $[M] := \{1, \cdots, M\}$. For random vector $X$, we use $p_X$ to denote its distribution or probability density function, depending on the context. For any vector $x \in \mathbb{R}^d$, we denote $\|\cdot\|_p$ as its $p$-norm, i.e., $\|x\|_p := (\sum_{i=1}^d |x_i|^p)^{1/p}$, and write $\|x\|_\infty := \max_i |x_i|$. We use $\|\cdot\|$ to denote the 2-norm for simplicity. For any vector $x \in \mathbb{R}^d$ and any $i, j \in [d]$ with $i < j$, we denote by $x_{i:j} \in \mathbb{R}^{j-i+1}$ the subvector consisting of the $i$-th through $j$-th entries of $x$. For vectors $\{\alpha_i\}_{i=1}^k$, we denote by

span($\{\alpha_i\}_{i=1}^k$) the linear space spanned by these vectors.

For a probability distribution $p$ and a random vector $X$, we write $X \sim p$ to mean that $X$ follows the distribution $p$. We denote by supp($p$) the support of probability measure $p$, i.e., the smallest closed set $S$ such that $p(S) = 1$. In addition, let $\mathbb{1}\{\cdot\}$ denote the indicator function. For random vectors $X, Y$, we define the 1-Wasserstein distance between their distributions $p_X$ and $p_Y$ by

$$W_1(p_X, p_Y) := \inf_{\gamma \in \Gamma(p_X, p_Y)} \iint \|x - y\| \, \gamma(\mathrm{d}x, \mathrm{d}y),$$

where $\Gamma(p_X, p_Y)$ denotes the set of couplings of $p_X$ and $p_Y$. For probability distributions $P, Q$, we denote their convolution by $P * Q$. Finally, we use poly($n$) to denote a polynomial function of $n$ where the specific degree may vary across different contexts.

## 2. Problem formulation

### 2.1. Preliminaries

In this section, we briefly introduce the score-based diffusion models.

**Forward process.** The forward process starts from the target distribution $p^\star$ and gradually adds Gaussian noise. A popular choice is the Ornstein-Uhlenbeck (OU) process (Song et al., 2020):

$$\mathrm{d}X_t = -X_t \, \mathrm{d}t + \sqrt{2} \, \mathrm{d}B_t, \quad \text{with} \quad X_0 \sim p^\star. \quad (2)$$

Here $(B_t)_{t \in [0,T]}$ is a standard Brownian motion in $\mathbb{R}^d$. A key property of this OU process is that the conditional distribution of $X_t$ given $X_0$ remains Gaussian for all $t$. More precisely, one can verify that

$$X_t \mid X_0 \overset{\mathrm{d}}{=} c_t X_0 + \sigma_t W_t \quad (3)$$

where $c_t := e^{-t}$, $\sigma_t := \sqrt{1 - e^{-2t}}$, and $W_t \sim \mathcal{N}(0, I_d)$ is independent of $X_0$. In particular, the parameter $t$ fully determines the noise level of the forward process. As $t$ becomes sufficiently large, $c_t$ approaches zero and the distribution of $X_t$ becomes close to the standard Gaussian distribution $\mathcal{N}(0, I_d)$.

**Reverse process.** Running the forward dynamics backward in time transforms Gaussian noise into samples from $p^\star$, forming the basis of diffusion-based sampling. For the OU process in (2), its time-reversal SDE is given by

$$Y_0 \sim p_{X_T},$$
$$\mathrm{d}Y_t = \left(Y_t + 2\nabla \log p_{X_{T-t}}(Y_t)\right) \mathrm{d}t + \sqrt{2} \, \mathrm{d}\overline{B}_t. \quad (4)$$

Here $p_{X_{T-t}}$ denotes the density of the forward process (2) at time $T-t$ and $\{\overline{B}_t\}_{t \in [0,T]}$ is a standard Brownian motion in

---

**Algorithm 1** Sampling procedure

---

1: **Input:** Early stopping time $\tau > 0$, end time $T > 0$, score estimator $\widehat{s}_{X_t}$ for $t \in [\tau, T]$.
2: Sample $y \sim \mathcal{N}(0, I_d)$.
3: Solve the reverse SDE:

$$\mathrm{d}\widehat{Y}_t = \left(\widehat{Y}_t + 2\widehat{s}_{X_{T-t}}(\widehat{Y}_t)\right) \mathrm{d}t + \sqrt{2}\, \mathrm{d}B_t \quad (6)$$

for $t \in [0, T - \tau]$ with $\widehat{Y}_0 = y$.
4: **Output:** generated sample $\widehat{Y}_{T-\tau}$.

---

$\mathbb{R}^d$. By classical time-reversal results for SDEs (Anderson, 1982), this process satisfies $Y_{T-t} \stackrel{\mathrm{d}}{=} X_t$ for all $t \in [0, T]$.

The crucial ingredient in the reverse dynamics is the score function of the marginal distributions of the forward process. For a random vector $X \in \mathbb{R}^d$ with density $p_X$, its *score function* is given by

$$s_X^\star(x) := \nabla \log p_X(x) = \frac{\nabla p_X(x)}{p_X(x)}. \quad (5)$$

Since these scores are unknown in practice, they must be estimated from training samples $\{X^{(i)}\}_{i=1}^n$ drawn from $p^\star$.

**Sampling procedure.** Since $X_T \stackrel{\mathrm{d}}{\to} \mathcal{N}(0, I_d)$ as $T \to \infty$, diffusion-based sampling can be implemented by initializing the reverse dynamics from $\mathcal{N}(0, I_d)$ and replacing the true score $s_{X_t}^\star$ with a learned estimator $\widehat{s}_{X_t}$. The resulting procedure is summarized in Algorithm 1. We introduce an early stopping time $\tau > 0$ to avoid the small-time regime, where score estimation is most challenging. In practice, the reverse SDE can be implemented using numerical methods such as Euler-Maruyama.

**Score estimation reduction.** To estimate the score function $s_{X_t}^\star$, it is often more convenient to construct score estimator for the following variance-exploding (VE) process

$$\mathrm{d}Z_t = \mathrm{d}B_t, \quad \text{with } Z_0 \sim p^\star. \quad (7)$$

Here $\{B_t\}_{t \geq 0}$ also denotes the Brownian motion and thus $Z_t$ follows the distribution $p^\star * \mathcal{N}(0, tI_d)$. It is straightforward to verify that the score functions of $X_t$ and $Z_t$ satisfy

$$s_{X_t}^\star(x) = \frac{1}{c_t} s_{Z_{h(t)}}^\star\left(\frac{x}{c_t}\right) \quad \text{with} \quad h(t) := \frac{\sigma_t^2}{c_t^2}. \quad (8)$$

As a result, it suffices to estimate the score function of $Z_t$ for any $t > 0$ and then define $\widehat{s}_{X_t}(x) := \frac{1}{c_t}\widehat{s}_{Z_{h(t)}}\left(\frac{x}{c_t}\right)$ as the estimator of $s_{X_t}^\star(x)$. For notational simplicity, we denote by $s_t^\star := \nabla \log p_{Z_t}$ the score function of $Z_t$, and let $\widehat{s}_t$ denote its estimator. The derivation of (8) is provided in Appendix A.2.

## 2.2. Assumptions

In this section, we introduce the assumptions imposed on the target distribution $p^\star$.

First, to capture low-dimensional, multi-modal structure, we assume that the support of $p^\star$ is contained in a finite union of low-dimensional linear subspaces.

**Assumption 1** (Union of low-dimensional subspaces). There exist linear subspaces $V_1, V_2, \ldots, V_M \subseteq \mathbb{R}^d$, with dimension $\dim(V_i) = k_i$, such that

$$\mathsf{supp}(p^\star) \subseteq \cup_{i=1}^M V_i.$$

Moreover, $p^\star$ assigns zero probability to intersections between different subspaces, i.e.,

$$p^\star(V_i \cap V_j) = 0, \quad \forall i \neq j. \quad (9)$$

Finally, each subspace has non-trivial mass:

$$p^\star(V_i) \geq \frac{1}{c_p M}, \quad \forall i \in [M] \quad (10)$$

for some constant $c_p > 0$.

This assumption provides a tractable abstraction for low-dimensional, multi-modal distributions, where different modes may concentrate on different subspaces. Such union-of-subspaces structure has been widely used in the modeling of heterogeneous high-dimensional data (Wang et al., 2024) and has also been observed empirically in real-world datasets (Brown et al., 2022; Kamkari et al., 2024).

For each $i \in [M]$, let $p_i^\star := p^\star |_{V_i}$ denote the restriction of the target distribution $p^\star$ to subspace $V_i$. By Assumption 1, we can decompose the target distribution as $p^\star = \sum_{i=1}^M p_i^\star$.

For each subspace $V_i$, let $A_i \in \mathbb{R}^{d \times k_i}$ be a matrix whose columns form an orthogonal basis of $V_i$:

$$V_i = \mathsf{span}(\mathsf{col}(A_i)), \quad A_i^\top A_i = I_{k_i}.$$

Denote by $\mathsf{proj}_i : \mathbb{R}^d \to V_i$ the projection onto $V_i$, given by

$$\mathsf{proj}_i(x) = A_i A_i^\top x.$$

*Remark* 1. Our framework can be extended naturally to distributions concentrated near a union of low-dimensional subspaces. In this paper, we focus on the noiseless setting to isolate the essential roles of low-dimensional structure and multi-modality, without introducing the additional technical complications caused by ambient noise. Extensions to noisy settings are discussed in Section 6.

Next, we impose a mild subgaussian assumption on the target distribution within each subspace.

**Assumption 2** (Subgaussian within each subspace). Let $p_i^{\text{low}}$ be the normalized push-forward distribution of $p_i^\star$ onto $\mathbb{R}^{k_i}$ under $A_i^\top$:

$$p_i^{\text{low}} := \mathsf{law}(A_i^\top Z), \quad Z \sim \frac{p_i^\star}{p^\star(V_i)}. \qquad (11)$$

We assume that $p_i^{\text{low}}$ is $\sigma_i$-subgaussian, that is, for any unit vector $\theta \in \mathbb{R}^{k_i}$ with $\|\theta\|_2 = 1$,

$$\mathbb{E}\big[\exp\big((X^\top \theta / \sigma_i)^2\big)\big] \leq 2, \quad X \sim p_i^{\text{low}}.$$

We denote $\sigma := \max_{i \in [M]} \sigma_i$.

The subgaussian assumption is fairly mild in the sense that it subsumes any distribution with bounded support, which covers a wide range of practical data such as image data.

## 3. Main results

This section introduces our score estimator and presents theoretical guarantees for both score estimation and sampling.

### 3.1. Algorithm

Given $n$ i.i.d. samples $\{X^{(i)}\}_{i=1}^n$ drawn from the target distribution $p^\star$, our goal is to build a score estimator $\widehat{s}_t$ that learns the score function $s_t^\star$ of the perturbed data distribution $p_t = p^\star * \mathcal{N}(0, tI_d)$ for any time $t > 0$.

**Motivation.** Observe that the density $p_t$ can be written as

$$p_t(x) = \big(p^\star * \mathcal{N}(0, tI_d)\big)(x) = \int \varphi_t(x - y; d) p^\star(\mathrm{d}y),$$

where $\varphi_t(x; d) := (2\pi t)^{-d/2} \exp\big(-\|x\|_2^2/(2t)\big)$ is the density of $\mathcal{N}(0, tI_d)$ in $\mathbb{R}^d$. Recall that under the UoS assumption, we can decompose $p^\star = \sum_{i=1}^M p_i^\star$, yielding

$$p_t(x) = \sum_{i=1}^M \int_{V_i} \varphi_t(x - y; d) p_i^\star(\mathrm{d}y).$$

The gradient of $p_t$ can then be computed as

$$\nabla p_t(x) = \sum_{i=1}^M \int_{V_i} -\frac{x - y}{t} \varphi_t(x - y; d) p_i^\star(\mathrm{d}y).$$

Therefore, the score function $s_t^\star = \nabla p_t / p_t$ of $p_t$ admits the following mixture-type decomposition:

$$s_t^\star(x) = \sum_{i=1}^M \frac{1}{p_t(x)} \int_{V_i} -\frac{x - y}{t} \varphi_t(x - y; d) p_i^\star(\mathrm{d}y)$$

$$=: \sum_{i=1}^M w_t(i, x) \cdot s_t(i, x), \qquad (12a)$$

where we define

$$w_t(i, x) := \frac{\int_{V_i} \varphi_t(x - y; d) p_i^\star(\mathrm{d}y)}{p_t(x)} =: \frac{q_t(i, x)}{p_t(x)}, \qquad (12b)$$

$$s_t(i, x) := \frac{\int_{V_i} -\frac{x-y}{t} \varphi_t(x - y; d) p_i^\star(\mathrm{d}y)}{\int_{V_i} \varphi_t(x - y; d) p_i^\star(\mathrm{d}y)}. \qquad (12c)$$

Intuitively, $w_t(i, x)$ can be interpreted as the posterior probability (or effective mixture weight) that $x$ originates from the $i$-th subspace after Gaussian smoothing, while $s_t(i, x)$ is the score of the corresponding smoothed component.

Our key observation is that each score component $s_t(i, x) \in \mathbb{R}^d$ admits a favorable normal-tangent decomposition. The normal part is essentially the score of a time-dependent Gaussian distribution and has a closed-form expression, while the tangent component is determined entirely by a $k_i$-dimensional score function (Chen et al., 2023):

$$s_t(i, x) = -\frac{1}{t}\big(x - \mathsf{proj}_i(x)\big) + A_i s_t^{\text{low}}(i, A_i^\top x), \qquad (13)$$

where $\mathsf{proj}_i(x) := A_i A_i^\top x$ is the projection of $x$ onto subspace $V_i$, and $s_t^{\text{low}}(i, \cdot) : \mathbb{R}^{k_i} \to \mathbb{R}^{k_i}$ is the score function of the $k_i$-dimensional smoothed distribution $p_i^{\text{low}} * \mathcal{N}(0, tI_{k_i})$ (see (11)) on the subspace $V_i$. This decomposition reduces score estimation to a low-dimensional problem: once $V_i$ is identified, estimating $s_t(i, \cdot)$ is governed by the difficulty of estimating the low-dimensional score $s_t^{\text{low}}(i, \cdot)$ in dimension $k_i$, rather than the ambient dimension $d$.

Motivated by this observation, we propose a two-step score estimation procedure. We first use the data to estimate the subspaces $\{A_i\}_{i=1}^M$ and construct a classification function $c : \mathbb{R}^d \to [M]$ that assigns points to subspaces (such that $c(x) = i$ if and only if $x \in V_i$). Given these estimates, we then estimate the component scores and mixture weights, and combine them to form the full score estimator. For theoretical clarity, we employ sample splitting, where $n_0$ samples are used for subspace recovery and the remaining $N = n - n_0$ samples are used for score estimation.

In what follows, we describe the proposed score estimator in reverse order.

**Score estimator.** We begin by presenting the score estimator assuming access to subspace estimates $\{A_i\}_{i=1}^M$ and a classification function $c(\cdot)$.

Inspired by the score decomposition in (12), we construct the score estimator as a weighted combination of score components associated with each subspace:

$$\widehat{s}_t(x) := \sum_{i=1}^M \widehat{w}_t(i, x) \widehat{s}_t(i, x), \qquad (14)$$

where $\widehat{w}_t(i, x)$ and $\widehat{s}_t(i, x)$ estimate $w_t(i, x)$ in (12b) and $s_t(i, x)$ in (12c), respectively.

- *Score component estimator* $\widehat{s}_t(i, x)$. In light of the low-dimensional structure in (13), it suffices to learn an estimator $\widehat{s}_t^{\mathsf{low}}$ for each $k_i$-dimensional score $s_t^{\mathsf{low}}(i, \cdot)$, and then construct the estimator for the $d$-dimensional score $s_t(i, x)$ associated with $V_i$ as

$$\widehat{s}_t(i, x) := -\frac{x - \mathsf{proj}_i(x)}{t} + A_i \widehat{s}_t^{\mathsf{low}}(i, A_i^\top x). \quad (15)$$

To build $\widehat{s}_t^{\mathsf{low}}$, let $\mathcal{C}_i := \{j \in [N] : X^{(j)} \in V_i\}$ denote the index set of samples belonging to subspace $V_i$. Since $s_t^{\mathsf{low}}(i, \cdot)$ is the score of the smoothed low-dimensional distribution $p_i^{\mathsf{low}} * \mathcal{N}(0, tI_{k_i})$, we first estimate its density using the Gaussian kernel estimator

$$\widehat{g}_t(i, x) := \frac{1}{|\mathcal{C}_i|} \sum_{j \in \mathcal{C}_i} \varphi_t(x - A_i^\top X^{(j)}; k_i), \quad (16)$$

where $\varphi_t(x; k_i)$ denotes the density of $\mathcal{N}(0, tI_{k_i})$. We then define the $k_i$-dimensional score estimator as

$$\widehat{s}_t^{\mathsf{low}}(i, x) := \mathsf{clip}_R\left( \frac{\nabla \widehat{g}_t(i, x)}{\widehat{g}_t(i, x)} \psi\left( \widehat{g}_t(i, x); \frac{\log N}{N(2\pi t)^{k_i/2}} \right) \right). \quad (17)$$

Here, $\psi(x; \eta) := \mathbb{1}\{x \geq \eta\}$ is a thresholding function, and the clip operator is defined by

$$\mathsf{clip}_r(z) := \begin{cases} z, & \|z\|_2 \leq r; \\ \frac{z}{\|z\|} \cdot r, & \text{otherwise.} \end{cases}$$

We set the clipping radius to be $R = \sqrt{2 \log N/t}$.

In words, we first form the plug-in estimator $\nabla \widehat{g}_t/\widehat{g}_t$ using the kernel density estimator (16). We then apply a thresholding rule $\psi(\widehat{g}_t; \eta_t)$, which regularizes this ratio according to the estimated density level $\widehat{g}_t$ and the threshold $\eta_t = N^{-1}(2\pi t)^{-k_i/2} \log N$ that depends on the sample size $N$ and time $t$. Specifically, in low-density regions where $\nabla \widehat{g}_t/\widehat{g}_t$ is unstable due to small denominators and limited data, the resulting score estimator $\widehat{s}_t$ is set to zero. This regularization is important not only for controlling the subsequent estimation error, but also for improving generalization by preventing the estimator from closely fitting empirical artifacts.

- *Weight estimator* $\widehat{w}_t(i, x)$. As for the mixture weight $w_t(i, x)$ defined in (12b), we first construct Gaussian kernel density estimators for $p_t(x)$ and $q_t(i, x)$:

$$\widehat{p}_t(x) := \frac{1}{N} \sum_{j=1}^{N} \varphi_t(x - X^{(j)}; d), \quad (18a)$$

and for any $i \in [M]$,

$$\widehat{q}_t(i, x) := \frac{1}{N} \sum_{j=1}^{N} \varphi_t(x - X^{(j)}; d) \mathbb{1}_{\{c(X^{(j)})=i\}}. \quad (18b)$$

We then define the weight estimator as

$$\widehat{w}_t(i, x) := \frac{\widehat{q}_t(i, x)}{\widehat{p}_t(x)} \mathbb{1}_{\{x \in \mathcal{G}_t(i)\}}, \quad (19)$$

where $\mathcal{G}_t(i)$ is a set given by

$$\mathcal{G}_t(i) := \left\{ x : \|x - \mathsf{proj}_i(x)\|_2 \leq R_t(i) \right\}, \quad (20)$$

with $R_t(i) = C_R \sqrt{td \log(Ndt^{k_i/2})}$ for some universal constant $C_R > 0$.

In a word, the weight estimator $\widehat{w}_t(i, x)$ is a plug-in estimator for the true weight (12b), up to some low-probability set under $p_t$. The indicator function $\mathbb{1}_{\{x \in \mathcal{G}_t(i)\}}$ is introduced for technical convenience in the analysis and could be removed with a shaper argument.

**Subspace recovery.** Finally, we briefly discuss how the subspace estimates $\{A_i\}_{i=1}^{M}$ and classification function $c(\cdot)$ can be obtained from training data. This is a classical subspace clustering problem. Under standard identifiability and separation conditions, and assuming known bounds on the number of subspaces $M$ and the maximal intrinsic dimension $k$, several polynomial-time methods can recover the underlying subspaces and cluster assignments, such as sparse subspace clustering (Elhamifar & Vidal, 2013), thresholding-based subspace clustering (Heckel & Bölcskei, 2015) and greedy subspace clustering (Park et al., 2014). From a statistical perspective, this geometric recovery step is typically less demanding than learning the full target distribution.

### 3.2. Theoretical guarantees

We now state our theoretical guarantees for the proposed score estimator and the resulting sampler.

We first present the $L^2$ error for the proposed score estimator $\widehat{s}_t$ in (14). The proof can be found in Appendix A.1.

**Theorem 1.** *Suppose the target distribution $p^\star$ satisfies Assumptions 1 and 2. Under the event of exact subspace recovery and $t \leq N^{O(1)}$, the $L^2$-error of the score estimator in (14) using $N$ samples satisfies*

$$\mathbb{E}\left[ \|\widehat{s}_t(X) - s_t^\star(X)\|_2^2 \right]$$
$$\leq C_{\mathsf{score}} \frac{dM^3}{N} \left( \frac{1}{t} + \frac{\sigma^{k \vee 2}}{t^{(k \vee 2)/2 + 1}} \right) \mathsf{poly} \log N$$

*for some constant $C_{\text{score}} > 0$ independent of $N$, $d$, $M$ and $t$. The expectation here is taken over the i.i.d. training samples $\{X^{(i)}\}_{i=1}^N$ used for score estimation and $X \sim p_t$.*

In words, Theorem 1 shows that the convergence rate of the $L^2$ error (with respect to diffusion time $t$) of the proposed score estimator depends on the intrinsic dimension $k$, rather than the ambient dimension $d$. This yields a substantial improvement over existing rate-optimal score estimation guarantees for general distributions (Wibisono et al., 2024; Zhang et al., 2024; Dou et al., 2024; Cai & Li, 2025),which do not exploit intrinsic low-dimensional data structures and therefore suffer from the curse of dimensionality.

Moreover, we note that exact subspace recovery can be achieved with high probability using $n_0 = C_{\text{sc}} M^2 k \log n$ samples for subspace clustering, for a sufficiently large constant $C_{\text{sc}} > 0$. This sample size is negligible compared with the remaining $N = n - n_0$ samples used for score estimation, provided that $n$ is sufficiently large.

We next translate the resulting score estimation guarantee into a sampling guarantee for the diffusion sampler. The proof is deferred to Appendix A.2.

**Theorem 2.** *Suppose the target distribution $p^\star$ satisfies Assumptions 1 and 2. Let $n_0 = C_{\text{sc}} M^2 k \log n$ for some large constant $C_{\text{sc}} > 0$ and $N = n - n_0$. Then for sufficiently large $n$, the output $\widehat{Y}_{T-\tau}$ of Algorithm 1, using the score estimator in (14) constructed from $N$ samples with $T = \log n$ and $\tau = n^{-2/k}$, satisfies*

$$\mathbb{E}\big[W_1(p^\star, p_{\widehat{Y}_{T-\tau}})\big] \le C d M^{3/2} n^{-\frac{1}{k \vee 2}} \operatorname{poly} \log n \quad (21)$$

*for some constant $C > 0$ independent of $n$, $d$ and $M$. Here the expectation is taken over the samples $\{X^{(i)}\}_{i=1}^n$.*

Theorem 2 provides the convergence rate of the $W_1$-sampling error for diffusion sampling equipped with the proposed score estimator. By exploiting the intrinsic low-dimensional data structure through kernel-based score estimation, the resulting convergence rate (with respect to sample size $n$) is governed by the intrinsic dimension $k$, rather than the ambient dimension $d$, with $d$ appearing only linearly through the prefactor.

Several remarks are in order:

- *(Near-)minimax optimality.* The minimax risk of estimating a $k$-dimensional density scales as $n^{-\frac{1}{k \vee 2}}$ (Chewi et al., 2024). Since sampling is always harder than density estimation, Theorem 2 shows that our sampling algorithm is minimax optimal (up to logarithmic factors).

- *Sample complexity.* The error bound in (21) demonstrates that in order to achieve an $\varepsilon$-accurate sampling

in 1-Wasserstein distance, it suffices to have $\varepsilon^{-(k \vee 2)}$ samples, up to logarithmic factors, thereby breaking the curse of dimensionality that plagues prior results (Wibisono et al., 2024; Zhang et al., 2024; Dou et al., 2024; Cai & Li, 2025).

- *Weak assumptions on the target distribution.* Our results do not rely on stringent structural conditions commonly imposed in earlier work, such as smooth densities/scores, log-concavity, or exactly Gaussian components. As a consequence, our framework applies to a broader class of multi-modal distributions of practical interest. Moreover, to the best of our knowledge, even in the single low-dimensional setting, our result is the first to achieve a (near-)optimal rate under only a sub-gaussian assumption on the target distribution, without extra assumptions on the score or density.

*Remark* 2. We believe that the linear dependence on $d$ in the prefactor of (21) is likely a proof artifact and may be improved through a sharper analysis. Determining whether this dependence is intrinsic or can be removed is an interesting direction for future work.

*Remark* 3. Prior theory (Cai & Li, 2025) suggests that, once the score estimation error is controlled, the discretization error of practical diffusion samplers does not affect the final statistical rate of the sampling error. Accordingly, this work focuses on the main statistical bottleneck, namely score estimation, by analyzing the idealized continuous-time reverse process. Meanwhile, establishing sharp Wasserstein discretization bounds under mild distributional conditions remains an important direction for future work.

## 4. Analysis

This section provides the proof sketches for Theorems 1–2.

**Proof sketch of Theorem 1.** In light of the expressions of the true score (12) and the score estimator (14), the $L^2$-error decomposes as:

$$\mathbb{E}\big[\|\widehat{s}_t(Z_t) - s_t^\star(Z_t)\|_2^2\big]$$
$$\lesssim \sum_{i=1}^M \int \mathbb{E}\big[\big(w_t(i, x) - \widehat{w}_t(i, x)\big)^2 \|\widehat{s}_t(i, x)\|_2^2\big] p_t(x) \mathrm{d}x$$
$$+ \sum_{i=1}^M \int w_t^2(i, x) \, \mathbb{E}\big[\|s_t(i, x) - \widehat{s}_t(i, x)\|_2^2\big] p_t(x) \mathrm{d}x, \quad (22)$$

where the expectation in the first line is taken over both $Z_t \sim p_t$ and the i.i.d. samples $\{X^{(i)}\}_{i=1}^N$, while the second and the third line only take expectation over the samples.

This decomposition suggests that we need to control the mean squared error of both the weight estimator $\widehat{w}_t(i, x)$ and score estimator $\widehat{s}_t(i, x)$.

For the weight estimator, we recall the true weight $w_t(i, x) = q_t(i, x)/p_t(x)$ in (12b). The mean squared error is controlled in the following lemma, with proof deferred to Appendix B.2.

**Lemma 1.** *For any $x \in \mathcal{G}_t(i)$, the weight estimator in (19) satisfies*

$$\mathbb{E}\big[\big(w_t(i, x) - \widehat{w}_t(i, x)\big)^2\big]$$
$$\lesssim \frac{1}{p_t^2(x)} \frac{1}{t^{d/2} N} \Big(\sum_{i=1}^{M} e^{-\frac{1}{2t}\|x - \mathsf{proj}_i(x)\|_2^2} q_t(i, x)\Big),$$

*where the expectation is taken over the i.i.d. samples $\{X^{(i)}\}_{i=1}^{N}$.*

*Remark* 4. The above mean squared error bound depends on the point $x$, density $p_t(x)$ and the geometric structure. This enables us to obtain a tight $L^2$ error bound and further get rid of the dependence on ambient dimension $d$.

With Lemma 1 in hand, we can now bound the first term in (22) associated with weight estimation. It is easy to see $p_t$ is $\sqrt{\sigma^2 + t}$-subgaussian. Define

$$\mathcal{B}_t := \big\{x \in \mathbb{R}^d : \|A_i^\top x\|_2 \le B_t, \forall i \in [M]\big\},$$

with $B_t := C_B \sqrt{k(\sigma^2 + t) \log N}$ for some universal constant $C_B > 0$. One can show that the estimation error within $\mathcal{B}_t$ dominates since $\mathcal{B}_t^c$ is a low probability region w.r.t. $p_t$. This allows us to apply Lemma 1 to derive the following bound that only depends on $k_i$:

$$\int \mathbb{E}\big[(w_t(i, x) - \widehat{w}_t(i, x))^2\big] p_t(x) \, \mathrm{d}x$$
$$\lesssim \sum_{j=1}^{M} \int_{\mathcal{B}_t} \frac{1}{N t^{d/2}} e^{-\frac{1}{2t}\|x - A_j A_j^\top x\|_2^2} \, \mathrm{d}x$$
$$= \widetilde{O}\Big(\sum_{j=1}^{M} \frac{(\sigma^2 + t)^{k_j/2}}{N t^{k_j/2}}\Big). \tag{23}$$

Regarding the second term in (22) associated with score estimation, notice that $w_t(i, x) = q_t(i, x)/p_t(x) \le 1$. Thus, it suffices to bound

$$\int w_t^2(i, x) \mathbb{E}\big[\|s_t(i, x) - \widehat{s}_t(i, x)\|_2^2\big] p_t(x) \, \mathrm{d}x$$
$$\le \int \mathbb{E}\big[\|s_t(i, x) - \widehat{s}_t(i, x)\|_2^2\big] q_t(i, x) \, \mathrm{d}x.$$

To this end, let $N_i := \sum_{j=1}^{N} \mathbb{1}_{\{c(X^{(j)})=i\}}$ denote the sample size that we use to estimate the score on subspace $V_i$. The following lemma provides a mean squared error bound for the score estimator $\widehat{s}_t(i, x)$ in (15). The proof is deferred to Appendix B.4.

**Lemma 2.** *For any fixed $n_i$, the score estimator $\widehat{s}_t(i, x)$ in (15) satisfies*

$$\int_{\mathbb{R}^d} \mathbb{E}\big[\|\widehat{s}_t(i, x) - s_t(i, x)\|_2^2 \mathbb{1}_{\{N_i \ge n_i\}}\big] q_t(i, x) \, \mathrm{d}x$$
$$\le C_{k_i} \frac{p_i^\star(V_i)}{n_i} \Big(\frac{1}{t} + \frac{\sigma^{k_i}}{t^{k_i/2+1}}\Big) \mathsf{poly} \log N \tag{24}$$

*for some constant $C_{k_i} > 0$ only depending on $k_i$. In addition, for any $x$, the $\ell_2$-norm of $\widehat{s}_t(i, x)$ is bounded by*

$$\|\widehat{s}_t(i, x)\|_2 \lesssim \frac{\|x - \mathsf{proj}_i(x)\|_2}{t} + \sqrt{\frac{2}{t} \log N}. \tag{25}$$

Combining (23)–(25) completes the proof of Theorem 1.

**Proof sketch of Theorem 2.** As we will show in Lemma 3 in Appendix B, $n_0 = C_{\mathsf{sc}} M^2 k \log n$ samples suffice to recover the subspaces exactly with probability at least $1 - Mn^{-10}$, for a sufficiently large constant $C_{\mathsf{sc}} > 0$. Thus, for large $n$, the remaining sample size for score estimation satisfies $N = n - n_0 \ge n/2$. Conditioned on the exact subspace recovery event, we apply the score estimation error bound in Theorem 1. We then relate the 1-Wasserstein error between the target distribution $p^\star$ and the generated distribution $p_{\widehat{Y}_{T-\tau}}$ to the integral of the score estimation error over time via the following stability bound (Oko et al., 2023; Azangulov et al., 2024; Tang & Yang, 2024):

$$\mathbb{E}\big[W_1(p^\star, p_{\widehat{Y}_{T-\tau}})\big]$$
$$\lesssim \sqrt{d}\Big(\sqrt{\tau} + \delta + e^{-T} +$$
$$\sum_{j=0}^{L-1} \sigma_{T_{j+1}} \sqrt{\log \frac{1}{\delta} \int_{T_j}^{T_{j+1}} \mathbb{E}\big[\|\widehat{s}_{X_t}(X) - s_{X_t}^\star(X)\|_2^2\big] \, \mathrm{d}t}\Big) \tag{26}$$

for $0 < T_0 = \tau < T_1 < \cdots < T_L = T$ and any $\delta > 0$. Here the expectation is taken over the randomness of samples and $X \sim p_{X_t}$.

By Theorem 1, together with $N \ge n/2$ and the score relation in (8), one can show that

$$\int_{\mathbb{R}^d} \mathbb{E}\big[\|\widehat{s}_{X_t}(x) - s_{X_t}^\star(x)\|_2^2\big] p_{X_t}(x) \, \mathrm{d}x$$
$$= \widetilde{O}\Big(\frac{dM^3}{n}\Big(\frac{1}{h(t)} + \frac{\sigma^{k \vee 2}}{h(t)^{(k \vee 2)/2+1}}\Big)\Big).$$

Observe that $h(t) = 2e^{2t} = 2/c_t^2$, and thus

$$\int_{T_j}^{T_{j+1}} \int_{\mathbb{R}^d} \mathbb{E}\big[\|\widehat{s}_{X_t}(x) - s_{X_t}^\star(x)\|_2^2\big] p_{X_t}(x) \, \mathrm{d}x \, \mathrm{d}t$$
$$= \widetilde{O}\Big(\frac{dM^3}{n}\Big(\log \frac{h(T_{j+1})}{h(T_j)} + \frac{2\sigma^{k \vee 2}}{k \vee 2} \frac{1}{h(T_j)^{(k \vee 2)/2}}\Big)\Big).$$

We then choose a dyadic partition of the time interval by setting

$$T_{j+1} = 2T_j, \quad T \asymp \log n, \quad \tau \asymp n^{-\gamma}$$

for some $\gamma > 0$ that will be specified below. Plugging these into (26) yields

$$\sum_{j=0}^{L-1} \sqrt{(1 - e^{-4T_j}) \int_{T_j}^{T_{j+1}} \mathbb{E}\big[\|\widehat{s}_{X_t}(X) - s_{X_t}^\star(X)\|_2^2\big] \, \mathrm{d}t}$$

$$\lesssim \sqrt{d}\, M^{3/2} \, \mathrm{poly} \log n \begin{cases} \frac{1}{\sqrt{n}} \tau^{-\frac{k}{4} + \frac{1}{2}}, & k \geq 2 \\ \frac{1}{\sqrt{n}}, & k = 1 \end{cases}$$

Finally, taking $\delta = n^{-1}$ and $\tau = n^{-2/k}$, we conclude

$$\mathbb{E}\big[W_1(p^\star, p_{\widehat{Y}_{T-\tau}})\big] \lesssim \frac{dM^{3/2}}{n^{1/(k \vee 2)}} \mathrm{poly} \log n.$$

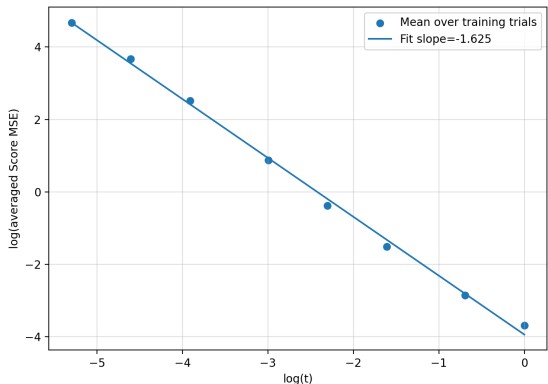

*Figure 1.* Empirical $L^2$-score error versus diffusion time $t$.

# 5. Numerical results

In this section, we provide numerical experiments to validate the theoretical findings of our paper. Since evaluating the Wasserstein distance is computationally prohibitive in high dimensions, we focus on the $L^2$-score estimation guarantee in Theorem 1.

We consider a synthetic target distribution in $\mathbb{R}^d$ with $d = 48$. The support is a union of $M = 128$ randomly generated linear subspaces, each with intrinsic dimension $k = 3$. The restriction of the distribution to each subspace is chosen to be a two-component Gaussian mixture with randomized parameters. We construct the kernel-based score estimator using $N = 50,000$ i.i.d samples from the target distribution. For each time value $t$, we approximate the $L^2$-score estimation error by Monte Carlo using 10,000 independent samples from $p_t$ and average the result over 20 independent training datasets generated from the same target distribution.

Figure 1 plots the empirical $L^2$ score estimation error versus the diffusion time $t$. The observed scaling is consistent with the prediction of Theorem 1, where the score estimation error is governed by the intrinsic dimension of the data, rather than the ambient dimension. In particular, despite the relatively large ambient dimension $d = 48$, the empirical error exhibits a substantially milder dependence on $t$ than would be suggested by ambient-dimensional worst-case bounds.

# 6. Discussion

This paper has studied the sample complexity of diffusion models for learning distributions supported on a union of low-dimensional subspaces, a tractable model for low-dimensional, multi-modal data commonly observed in practice. We construct a kernel-based score estimator and prove that diffusion-based sampling can learn the target distribution using at most $\widetilde{O}(\varepsilon^{-(k \vee 2)})$ samples, where $k$ represents the intrinsic dimension. Our result shows that diffusion models can achieve statistical optimality by exploiting intrinsic low-dimensional structure while naturally accommodating multi-modal data.

Building on the results of our paper, several directions remain open for future work. First, it would be valuable to extend our theory to practical NN-based score estimators. A natural approach is to analyze an ERM estimator over a NN class, which requires controlling both approximation and generalization errors. The main challenge lies in the approximation step: constructing a NN approximation of the target score whose complexity depends on the intrinsic dimension. Our score decomposition, together with the kernel-based construction, makes the relevant low-dimensional approximation targets explicit. Thus, our construction provides a concrete roadmap for future analysis of NN score estimators. Second, it would be important to extend our framework to real data with more complex geometric structures, such as classes residing on manifolds of varying dimensions. A natural first step is to consider distributions concentrated near a union of low-dimensional subspaces. In this setting, once the underlying subspaces are learned from noisy observations, the score still admits an analogous normal-tangent decomposition, where the tangent component is governed by a low-dimensional score and the normal component remains Gaussian with an enlarged variance. This suggests that the framework developed here could be extended to noisy low-dimensional models. Third, it would be interesting to develop a fully end-to-end convergence analysis that explicitly accounts for both score estimation error and discretization of the reverse-time dynamics (SDE/ODE), ideally yielding non-asymptotic bounds in Wasserstein distance under mild distributional conditions.

## Acknowledgements

C. Cai is supported in part by the NSF grant DMS-2515333.

## Impact Statement

This paper presents work whose goal is to advance the field of diffusion model theory. There are many potential societal consequences of our work, none of which we feel must be specifically highlighted here.

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

# A. Proof of theorems

## A.1. Proof of Theorem 1

In this section, we consider the score estimation error under exact subspace recovery. We first define the following event set $\mathcal{A}$ as

$$\mathcal{A} := \left\{ N_i \geq \frac{N}{2c_p M}, \quad \forall i \in [M] \right\}.$$

Here $N_i := \sum_{j=1}^{N} \mathbb{1}_{\{c(X^{(j)})=i\}}$ is a random number of samples on $V_i$. The following claim tells us that, event $\mathcal{A}$ happens with high probability.

**Claim 1.** *Under Assumption 1, it holds that,*

$$\mathbb{P}[\mathcal{A}^c] \leq M e^{-\frac{N}{2c_p^2 M^2}}.$$

The proof follows from concentration inequality and can be found in Appendix C.1.

We first consider the error on event $\mathcal{A}$, where we have enough samples on each subspace. Notice the score decomposition (12) and the score estimator in (14), the $L^2$ estimation error can be written as

$$\int \mathbb{E}\left[ \|s_t^\star(x) - \widehat{s}_t(x)\|_2^2 \mathbb{1}_{\mathcal{A}} \right] p_t(x)\mathrm{d}x = \int \mathbb{E}[\| \sum_{i=1}^{M} \left( w_t(i,x)s_t(i,x) - \widehat{w}_t(i,x)\widehat{s}_t(i,x) \right) \|_2^2 \mathbb{1}_{\mathcal{A}}] p_t(x)\mathrm{d}x$$

$$\leq M \sum_{i=1}^{M} \int \mathbb{E}\left[ \|w_t(i,x)s_t(i,x) - \widehat{w}_t(i,x)\widehat{s}_t(i,x)\|_2^2 \mathbb{1}_{\mathcal{A}} \right] p_t(x)\mathrm{d}x \quad \text{(C-S Ineq)}$$

$$=: M \sum_{i=1}^{M} L_i.$$

We further decompose $L_i$ using the error of weight estimator and the error of score estimator respectively,

$$L_i := \int \mathbb{E}\left[ \|w_t(i,x)s_t(i,x) - \widehat{w}_t(i,x)\widehat{s}_t(i,x)\|_2^2 \mathbb{1}_{\mathcal{A}} \right] p_t(x)\mathrm{d}x$$

$$\lesssim \int \mathbb{E}\left[ \left( w_t(i,x) - \widehat{w}_t(i,x) \right)^2 \cdot \|\widehat{s}_t(i,x)\|_2^2 \mathbb{1}_{\mathcal{A}} \right] p_t(x)\mathrm{d}x + \int \mathbb{E}\left[ \left( w_t(i,x) \right)^2 \cdot \|s_t(i,x) - \widehat{s}_t(i,x)\|_2^2 \mathbb{1}_{\mathcal{A}} \right] p_t(x)\mathrm{d}x$$

$$=: L_{i,1} + L_{i,2}.$$

**Bound of $L_{i,1}$.** We utilize a set $\mathcal{B}_t$ here and further decompose $L_{i,1}$ as

$$L_{i,1} := \int \mathbb{E}\left[ \left( w_t(i,x) - \widehat{w}_t(i,x) \right)^2 \cdot \|\widehat{s}_t(i,x)\|_2^2 \mathbb{1}_{\mathcal{A}} \right] p_t(x)\mathrm{d}x$$

$$\leq \int_{\mathcal{G}_t(i) \cap \mathcal{B}_t} \mathbb{E}\left[ \left( w_t(i,x) - \widehat{w}_t(i,x) \right)^2 \cdot \|\widehat{s}_t(i,x)\|_2^2 \mathbb{1}_{\mathcal{A}} \right] p_t(x)\mathrm{d}x$$

$$+ \int_{\mathcal{G}_t(i)^c \cap \mathcal{B}_t} \mathbb{E}\left[ \left( w_t(i,x) - \widehat{w}_t(i,x) \right)^2 \cdot \|\widehat{s}_t(i,x)\|_2^2 \mathbb{1}_{\mathcal{A}} \right] p_t(x)\mathrm{d}x$$

$$+ \int_{\mathcal{B}_t^c} \mathbb{E}\left[ \left( w_t(i,x) - \widehat{w}_t(i,x) \right)^2 \cdot \|\widehat{s}_t(i,x)\|_2^2 \mathbb{1}_{\mathcal{A}} \right] p_t(x)\mathrm{d}x$$

$$=: \kappa_1 + \kappa_2 + \kappa_3,$$

here we recall that $\mathcal{G}_t(i)$ is the regularization set defined in (20) and $\mathcal{B}_t$ is defined as,

$$\mathcal{B}_t := \left\{ x \in \mathbb{R}^d : \|A_i^\top x\|_2 \leq B_t, \forall i \in [M] \right\}, \quad \text{with } B_t := C_B \sqrt{k(\sigma^2 + t)\log(N)}. \tag{27}$$

for certain absolute constant $C_B > 0$.

The following claim tells us that the probability outside $\mathcal{B}_t$ is negligible and see Appendix C.2 for its proof.

**Claim 2.** *Under Assumption 2, for the set $\mathcal{B}_t$, it holds that,*

$$\int_{\mathcal{B}_t^c} p_t(x)\mathrm{d}x \lesssim \frac{Mk}{N^4}$$

$$\int_{\mathcal{B}_t^c} \|x\|_2^2 \cdot p_t(x)\mathrm{d}x \lesssim \frac{d\sqrt{M}}{N^4}(\sigma^2 + t).$$

- For $\kappa_1$, notice that for $x \in \mathcal{G}_t(i)$,

$$\|\widehat{s}_t(i,x)\|_2 \leq \frac{R_t(i)}{t} + \sqrt{\frac{2}{t}\log N}$$

using Lemma 2. Hence,

$$
\begin{aligned}
\kappa_1 &:= \int_{\mathcal{G}_t(i)\cap\mathcal{B}_t} \mathbb{E}\Big[\big(w_t(i,x) - \widehat{w}_t(i,x)\big)^2 \cdot \|\widehat{s}_t(i,x)\|_2^2 \mathbb{1}_{\mathcal{A}}\Big] p_t(x)\mathrm{d}x \\
&\lesssim \Big(\frac{R_t(i)^2}{t^2} + \frac{1}{t}\Big) \cdot \int_{\mathcal{G}_t(i)\cap\mathcal{B}_t} \frac{1}{(2\pi t)^{d/2} N \cdot p_t(x)}\Big(\sum_{j=1}^{M} e^{-\frac{1}{2t}\|x-\mathsf{proj}_j(x)\|_2^2} \cdot q_t(j,x)\Big)\mathrm{d}x \quad \text{(Lemma 1)} \\
&\leq \Big(\frac{R_t(i)^2}{t^2} + \frac{1}{t}\Big) \cdot \sum_{j=1}^{M} \int_{\mathcal{B}_t} \frac{1}{N(2\pi t)^{d/2}} e^{-\frac{1}{2t}\|x-A_j A_j^\top x\|_2^2}\mathrm{d}x \\
&\leq \Big(\frac{R_t(i)^2}{t^2} + \frac{1}{t}\Big) \cdot \sum_{j=1}^{M} \frac{2^{k_j} B_t^{k_j}}{N(2\pi t)^{k_j/2}} \\
&\lesssim \frac{d}{t} \cdot \frac{M}{N}\Big(1 + \frac{\sigma^k}{t^{k/2}}\Big) \cdot \big(\mathsf{poly}\log N + \log t\big)
\end{aligned}
$$

In the third inequality, we apply the following changing variable technique in integration and then use Tonelli's Theorem,

$$z = T_j x := \begin{pmatrix} A_j^\top x \\ P_{V_j^\perp}^\top x \end{pmatrix} \tag{28}$$

here columns of $P_{V_j^\perp}$ denotes an orthogonal basis of $V_j^\perp$ and this is an orthogonal transform with

$$\|x - A_j A_j^\top x\|_2^2 = \|P_{V_j^\perp}^\top x\|_2^2.$$

- For $\kappa_2$, we have,

$$
\begin{aligned}
\kappa_2 &:= \int_{\mathcal{G}_t(i)^c\cap\mathcal{B}_t} \mathbb{E}\Big[\big(w_t(i,x) - \widehat{w}_t(i,x)\big)^2 \cdot \|\widehat{s}_t(i,x)\|_2^2 \mathbb{1}_{\mathcal{A}}\Big] p_t(x)\mathrm{d}x \\
&= \int_{\mathcal{G}_t(i)^c\cap\mathcal{B}_t} \frac{q_t^2(i,x)}{p_t^2(x)}\mathbb{E}[\|\widehat{s}_t(i,x)\|_2^2] \cdot p_t(x)\mathrm{d}x \\
&\lesssim \int_{\mathcal{G}_t(i)^c\cap\mathcal{B}_t} \Big(\frac{1}{t} + \frac{\|x - A_i A_i^\top x\|_2^2}{t^2}\Big) \cdot q_t(i,x)\mathrm{d}x \quad \text{(Lemma 2 + } q_t(i,x) \leq p_t(x)\text{)}
\end{aligned}
$$

Since it holds that,

$$q_t(i,x) := \int_{V_i} \varphi_t(x - y; d)p_i^\star(\mathrm{d}y) \leq p_i^\star(V_i) \cdot (2\pi t)^{-d/2} e^{-\frac{1}{2t}\|x-A_i A_i^\top x\|_2^2},$$

we apply the similar linear transform as (28),

$$
\begin{aligned}
\kappa_2 &\lesssim p_i^\star(V_i) \int_{\mathbb{R}^d} \mathbb{1}_{\|z\| \leq B_t} \cdot \mathbb{1}_{\|z_{k_i+1:d}\|_2 \geq R_t(i)} \Big( \frac{1}{t} + \frac{\|z_{k_i+1:d}\|_2^2}{t^2} \Big) \cdot (2\pi t)^{-d/2} e^{-\frac{1}{2t}\|z_{k_i+1:d}\|_2^2} \mathrm{d}z \\
&\lesssim p_i^\star(V_i)(2\pi t)^{-k_i/2} \int_{\mathbb{R}^{k_i}} \mathbb{1}_{\|z_{1:k_i}\| \leq B_t} \Big( \frac{1}{t} + \frac{d \cdot R_t(i)^2 + d^2 t C_1}{t^2} \Big) \cdot \exp\Big( -\frac{R_t(i)^2}{d \cdot t C_1} \Big) \mathrm{d}z_{1:k_i} \quad \text{(Lemma 6)} \\
&\lesssim \frac{2^{k_i} B_t^{k_i}}{(2\pi t)^{k_i/2}} p_i^\star(V_i) \cdot \Big( \frac{d^2}{t} + \frac{d \cdot R_t(i)^2}{t^2} \Big) \exp\Big( -\frac{R_t(i)^2}{d \cdot t C_1} \Big) \\
&\lesssim \frac{d}{Nt}\Big( 1 + \frac{\sigma^{k_i}}{t^{k_i/2}} \Big) \cdot \mathsf{poly}\log N \lesssim \frac{d}{Nt}\Big( 1 + \frac{\sigma^k}{t^{k/2}} \Big) \cdot \mathsf{poly}\log N.
\end{aligned}
$$

Here $C_1 > 0$ is a universal constant which is related with the sub-gaussian norm of standard Gaussian distribution.

- For $\kappa_3$,

$$
\begin{aligned}
\kappa_3 &:= \int_{\mathcal{B}_t^c} \mathbb{E}\Big[ \big( w_t(i,x) - \widehat{w}_t(i,x) \big)^2 \cdot \|\widehat{s}_t(i,x)\|_2^2 \mathbb{1}_{\mathcal{A}} \Big] p_t(x) \mathrm{d}x \\
&\leq \int_{\mathcal{B}_t^c} \mathbb{E}\Big[ \|\widehat{s}_t(i,x)\|_2^2 \Big] p_t(x) \mathrm{d}x \\
&\lesssim \int_{\mathcal{B}_t^c} \Big( \frac{\|x - A_i A_i^\top x\|_2^2}{t^2} + \frac{1}{t} \Big) \cdot p_t(x) \mathrm{d}x \quad \text{(Lemma 2)} \\
&\leq \int_{\mathcal{B}_t^c} \Big( \frac{\|x\|_2^2}{t^2} + \frac{1}{t} \Big) p_t(x) \mathrm{d}x \\
&\lesssim \frac{dM(\sigma^2 + t)}{t^2 N^2} \mathsf{poly}\log N \quad \text{(Claim 2)} \\
&\lesssim \frac{d}{tN}\Big( 1 + \frac{\sigma^{k \vee 2}}{t^{(k \vee 2)/2}} \Big) \mathsf{poly}\log N.
\end{aligned}
$$

- For $L_{i,1}$, we could sum them up,

$$
\begin{aligned}
L_{i,1} &\leq \kappa_1 + \kappa_2 + \kappa_3 \\
&\lesssim \frac{dM}{Nt}\Big( 1 + \frac{\sigma^{k \vee 2}}{t^{(k \vee 2)/2}} \Big) \cdot \big( \mathsf{poly}\log N + \log t \big).
\end{aligned}
$$

**Bound of $L_{i,2}$.**

$$
\begin{aligned}
L_{i,2} &:= \int \mathbb{E}\Big[ \big( w_t(i,x) \big)^2 \cdot \|s_t(i,x) - \widehat{s}_t(i,x)\|_2^2 \cdot \mathbb{1}_{\mathcal{A}} \Big] p_t(x) \mathrm{d}x \\
&= \int \frac{q_t^2(i,x)}{p_t^2(x)} \mathbb{E}\Big[ \|s_t(i,x) - \widehat{s}_t(i,x)\|_2^2 \cdot \mathbb{1}_{\mathcal{A}} \Big] p_t(x) \mathrm{d}x \\
&\leq \int \mathbb{E}\Big[ \|s_t(i,x) - \widehat{s}_t(i,x)\|_2^2 \cdot \mathbb{1}_{\{N_i \geq N/2c_p M\}} \Big] q_t(i,x) \mathrm{d}x \quad \text{(since } q_t(i,x) \leq p_t(x)\text{)} \\
&\lesssim p^\star(V_i) \frac{c_p M (4/\sqrt{\pi})^{k_i}}{N} \Big( \frac{1}{t} + \frac{\sigma^{k_i}}{t^{k_i/2+1}} \Big) \mathsf{poly}\log N \quad \text{(Lemma 2)}.
\end{aligned}
$$

**Bound of error on $\mathcal{A}$.** Therefore,

$$
\begin{aligned}
\int \mathbb{E}[\|s_t^\star(x) - \widehat{s}_t(x)\|_2^2 \mathbb{1}_{\mathcal{A}}] p_t(x) \mathrm{d}x &\leq M \sum_{i=1}^{M} (L_{i,1} + L_{i,2}) \\
&\lesssim \frac{dM^3}{Nt}\Big( 1 + \frac{\sigma^{k \vee 2}}{t^{(k \vee 2)/2}} \Big) \cdot \big( \mathsf{poly}\log N + \log t \big).
\end{aligned}
$$

Here for simplicity, we omit constant terms $c_p$ and terms that are only related with intrinsic dimension $k$.

**Bound of error outside $\mathcal{A}^c$.**

$$\int \mathbb{E}[\|s_t^\star(x) - \widehat{s}_t(x)\|_2^2 \mathbb{1}_{\mathcal{A}^c}]p_t(x)\mathrm{d}x \lesssim \left(\int \|s_t(x)\|_2^2 p_t(x)\mathrm{d}x\right) \cdot \mathbb{P}[\mathcal{A}^c]$$
$$+ \int \left(\frac{1}{t} + \frac{\|x\|_2^2}{t^2}\right)p_t(x)\mathrm{d}x \cdot \mathbb{P}[\mathcal{A}^c] \quad \text{(Lemma 2)}$$
$$\lesssim \left(\frac{d}{t} + \frac{d(\sigma^2 + t)}{t^2}\right) \cdot 2Me^{-\frac{N}{2c_p^2 M^2}}$$

Here we apply Lemma 11 in (Cai & Li, 2025) and that $p^\star * \mathcal{N}(0, tI_d)$ is $c\sqrt{\sigma^2 + t}$ subgaussian r.v as we have proven in Appendix C.2. Notice that, for large enough $N$, $e^{-\frac{N}{2c_p^2 M^2}} \leq \frac{M^2 c_p^2}{N}$ and thus,

$$\int \mathbb{E}[\|s_t^\star(x) - \widehat{s}_t(x)\|_2^2 \mathbb{1}_{\mathcal{A}^c}]p_t(x)\mathrm{d}x \lesssim \frac{dM^3}{Nt}\left(1 + \frac{\sigma^2}{t}\right).$$

**Summary.** In summary, our analysis above shows that,

$$\int \mathbb{E}\big[\|\widehat{s}_t(x) - s_t^\star(x)\|_2^2\big]p_t(x)\mathrm{d}x = \int \mathbb{E}[\|s_t^\star(x) - \widehat{s}_t(x)\|_2^2 \mathbb{1}_{\mathcal{A}}]p_t(x)\mathrm{d}x + \int \mathbb{E}[\|s_t^\star(x) - \widehat{s}_t(x)\|_2^2 \mathbb{1}_{\mathcal{A}^c}]p_t(x)\mathrm{d}x$$
$$\lesssim \frac{dM^3}{Nt}\left(1 + \frac{\sigma^{k\vee 2}}{t^{(k\vee 2)/2}}\right) \cdot \big(\text{poly}\log N + \log t\big).$$

This proves Theorem 1.

### A.2. Proof of Theorem 2

**Proof of (8).** Recall the forward process (3), it holds that,

$$p_{X_t}(x) = \int p_{X_t|X_0}(x|y)p^\star(\mathrm{d}y)$$
$$= \int \left(2\pi\sigma_t^2\right)^{-d/2} e^{-\frac{1}{2\sigma_t^2}\|x - c_t y\|_2^2}p^\star(\mathrm{d}y)$$

Hence,

$$\nabla_x p_{X_t}(x) = (2\pi\sigma_t^2)^{-d/2} \cdot \int \left(-\frac{x - c_t y}{\sigma_t^2}\right)e^{-\frac{1}{2\sigma_t^2}\|x - c_t y\|_2^2}p^\star(\mathrm{d}y)$$

Therefore, its score function,

$$s_{X_t}^\star(x) = \frac{\nabla_x p_{X_t}(x)}{p_{X_t}(x)}$$
$$= \frac{\int \left(-\frac{x - c_t y}{\sigma_t^2}\right)e^{-\frac{1}{2\sigma_t^2}\|x - c_t y\|_2^2}p^\star(\mathrm{d}y)}{\int e^{-\frac{1}{2\sigma_t^2}\|x - c_t y\|_2^2}p^\star(\mathrm{d}y)}$$

Similarly, for any $t > 0$ and the VE process (7), it holds that,

$$s_{Z_t}^\star(x) = \frac{\int \left(-\frac{x - y}{t}\right)e^{-\frac{1}{2t}\|x - y\|_2^2}p^\star(\mathrm{d}y)}{\int e^{-\frac{1}{2t}\|x - y\|_2^2}p^\star(\mathrm{d}y)}$$

Therefore,

$$
\begin{aligned}
s_{X_t}^\star(c_t x) &= \frac{\int \left(-\frac{c_t(x-y)}{\sigma_t^2}\right) e^{-\frac{c_t^2}{2\sigma_t^2}\|x-y\|_2^2} p^\star(\mathrm{d}y)}{\int e^{-\frac{c_t^2}{2\sigma_t^2}\|x-y\|_2^2} p^\star(\mathrm{d}y)} \\
&= \frac{1}{c_t} s_{Z_{h(t)}}^\star(x) \\
&=: \frac{1}{c_t} s_{h(t)}^\star(x)
\end{aligned}
$$

Here $h(t) := \frac{\sigma_t^2}{c_t^2} = \frac{1-e^{-2t}}{e^{-2t}} = e^{2t} - 1$.

**Proof of Theorem 2.** We denote $\mathcal{E}$ as the event set of exact subspace recovery using $n_0$ samples. From Lemma 3, we could take $n_0 = O(M^2 k \log n)$ and thus $\mathbb{P}[\mathcal{E}^c] \lesssim Mn^{-10}$. Hence, for large $n$ such that $n_0 \leq 0.5n$, we have the sample size for score estimation: $N \geq 0.5n$. Therefore, for the score estimation error of VE process, we have,

$$
\begin{aligned}
\mathbb{E}\big[\|\widehat{s}_t(X) - s_t^\star(X)\|_2^2\big] &= \mathbb{E}\big[\|\widehat{s}_t(X) - s_t^\star(X)\|_2^2 \cdot \mathbb{1}_{\mathcal{E}}\big] + \mathbb{E}\big[\|\widehat{s}_t(X) - s_t^\star(X)\|_2^2 \cdot \mathbb{1}_{\mathcal{E}^c}\big] \\
&\lesssim \frac{dM^3}{nt}\left(1 + \frac{\sigma^{k \vee 2}}{t^{(k \vee 2)/2}}\right) \cdot (\text{poly}\log n + \log t) + \frac{d\log n}{t} \cdot Mn^{-10} + \sqrt{\mathbb{E}[\|s_t^\star(X)\|_2^4] \cdot Mn^{-10}} \\
&\lesssim \frac{dM^3}{nt}\left(1 + \frac{\sigma^{k \vee 2}}{t^{(k \vee 2)/2}}\right) \cdot (\text{poly}\log n + \log t) \quad\quad (29)
\end{aligned}
$$

In the first inequality, we apply Theorem 1 and that $\|\widehat{s}_t(x)\|^2 \lesssim \frac{R_t^2(i)}{t^2} + \frac{\log n}{t} \lesssim \frac{d\log n}{t}$; in the last inequality, we apply Lemma 11 in (Cai & Li, 2025) for the moment bound of true scores.

Now we consider the sampling error results from score estimation error. Notice that 1-Wasserstein distance between target distribution and generated distribution $\widehat{Y}_{T-\tau}$ using Algorithm 1 has the following control,

$$
\mathbb{E}\big[W_1(p^\star, p_{\widehat{Y}_{T-\tau}})\big] \lesssim \sqrt{d}\Big(\sqrt{\tau} + \sum_{j=0}^{L-1} \sqrt{\log \delta^{-1} \cdot \sigma_{T_{j+1}}^2 \int_{T_j}^{T_{j+1}} \int_{\mathbb{R}^d} \mathbb{E}\big[\|\widehat{s}_{X_t}(x) - s_{X_t}^\star(x)\|_2^2\big] p_{X_t}(x)\, \mathrm{d}x\, \mathrm{d}t} + \delta + e^{-T}\Big) \quad (30)
$$

for certain $0 < T_0 = \tau < T_1 < \cdots < T_L = T$ and any $\delta > 0$. This bound can be found as (8) in Azangulov et al. (2024), which cites Lemma D.7 in Oko et al. (2023) as the proof, and also can be found as Lemma B.2 in Tang & Yang (2024). Furthermore, the score estimator we define in (14) satisfies,

$$
\widehat{w}_t(i,x) \neq 0 \implies x \in \mathcal{G}_t(i) \implies \|\widehat{s}_t(i,x)\| \lesssim \sqrt{\frac{\log n}{t}}
$$

and hence, $\|\widehat{s}_t(x)\| \lesssim \sqrt{\frac{\log n}{t}}$, with,

$$
\widehat{s}_{X_t}(x) = \frac{1}{c_t}\widehat{s}_{h(t)}\left(\frac{x}{c_t}\right) \lesssim \frac{1}{c_t}\sqrt{\frac{\log n}{\sigma_t^2/c_t^2}} = \sqrt{\frac{\log n}{\sigma_t^2}}.
$$

Therefore, the condition for (30) is satisfied in our setting. We first derive a bound for $\int_{\mathbb{R}^d} \mathbb{E}\big[\|\widehat{s}_{X_t}(x) - s_{X_t}^\star(x)\|_2^2\big] p_{X_t}(x)\, \mathrm{d}x$ using the result in Theorem 1,

$$\int_{\mathbb{R}^d} \mathbb{E}\big[\|\widehat{s}_{X_t}(x) - s^\star_{X_t}(x)\|_2^2\big] p_{X_t}(x)\,\mathrm{d}x = \frac{1}{c_t^2} \int_{\mathbb{R}^d} \mathbb{E}\big[\|\widehat{s}_{h(t)}(\frac{x}{c_t}) - s^\star_{h(t)}(\frac{x}{c_t})\|_2^2\big] p_{X_t}(x)\,\mathrm{d}x \quad \text{(Equation (8))}$$

$$= \int \frac{1}{c_t^2}(c_t^2)^{d/2} \mathbb{E}\big[\|\widehat{s}_{h(t)}(y) - s^\star_{h(t)}(y)\|_2^2\big] p_{X_t}(c_t y)\,\mathrm{d}y$$

$$= \frac{1}{c_t^2} \int \mathbb{E}\big[\|\widehat{s}_{h(t)}(y) - s^\star_{h(t)}(y)\|_2^2\big] p_{Z_{h(t)}}(y)\,\mathrm{d}y$$

$$\lesssim \frac{1}{c_t^2} \frac{dM^3}{n}\Big(\frac{1}{h(t)} + \frac{\sigma^{k\vee 2}}{h(t)^{(k\vee 2)/2+1}}\Big)\big(\mathsf{poly}\log n + \log h(t)\big) \quad \text{(Equation (29))}.$$

Observe that,

$$h'(t) = 2e^{2t} = 2/c_t^2$$

and thus,

$$\int_{T_j}^{T_{j+1}} \int_{\mathbb{R}^d} \mathbb{E}\big[\|\widehat{s}_{X_t}(x) - s^\star_{X_t}(x)\|_2^2\big] p_{X_t}(x)\,\mathrm{d}x\,\mathrm{d}t \lesssim \frac{dM^3}{n}\Big(\int_{T_j}^{T_{j+1}} \Big(\frac{1}{h(t)} + \frac{\sigma^{k\vee 2}}{h(t)^{(k\vee 2)/2+1}}\Big)h'(t)\,\mathrm{d}t\Big)\big(\mathsf{poly}\log n + \log h(t)\big)$$

$$\lesssim \frac{dM^3}{n}\Big(\log\frac{h(T_{j+1})}{h(T_j)} + \frac{2\sigma^{k\vee 2}}{k\vee 2}\frac{1}{h(T_j)^{(k\vee 2)/2}}\Big)\big(\mathsf{poly}\log n + T\big).$$

We then take the synthesized discretization as,

$$T_{j+1} = 2T_j, \quad T \asymp \log n, \quad \tau \asymp n^{-\gamma}$$

for some $\gamma > 0$ that will be determined later. Then we can easily check that $L \asymp \log n$ and,

$$\sum_{j=0}^{L-1} \sqrt{(1 - e^{-4T_j}) \cdot \int_{T_j}^{T_{j+1}} \int_{\mathbb{R}^d} \mathbb{E}\big[\|\widehat{s}_{X_t}(x) - s^\star_{X_t}(x)\|_2^2\big] p_{X_t}(x)\,\mathrm{d}x\,\mathrm{d}t}$$

$$\lesssim \frac{\sqrt{d}M^{3/2}\mathsf{poly}\log n}{\sqrt{n}}\Big(\sum_{j=0}^{L-1} \sqrt{(1 - e^{-4T_j})\frac{2\sigma^{k\vee 2}}{k\vee 2}\frac{1}{(e^{2T_k}-1)^{(k\vee 2)/2}}} + \mathsf{poly}\log n\Big)$$

$$\lesssim \frac{\sqrt{d}M^{3/2}\mathsf{poly}\log n}{\sqrt{n}} \cdot \Big(\sum_{j=0}^{L-1} \sqrt{\frac{4\sigma^k T_j}{2^{k/2}T_j^{(k\vee 2)/2}}} + 1\Big)$$

$$\lesssim \begin{cases} \frac{\sqrt{d}M^{3/2}\mathsf{poly}\log n}{\sqrt{n}}\tau^{-\frac{k}{4}+\frac{1}{2}}, & k \geq 2 \\ \frac{\sqrt{d}M^{3/2}\mathsf{poly}\log n}{\sqrt{n}}, & k = 1 \end{cases}.$$

Further take $\delta = n^{-1}$ and take $\tau = n^{-2/k}$, then it holds that,

$$W_1(p^\star, p_{\widehat{Y}_{T-\tau}}) \lesssim \frac{dM^{3/2}\mathsf{poly}\log n}{n^{1/(k\vee 2)}}.$$

## B. Proof of Lemmas

As long as we could ensure that with high probability the data segmentation with $n_0$ points are correct and each class contains at least $k + 1$ points, then we could recover the exact linear subspaces $V_j$ and $c_j(X) = \mathbb{1}_{\{X \in V_j\}}$ almost surely for $X \sim p^\star$. This gives the following Lemma 3.

**Lemma 3** (Subspace Clustering). *Under Assumption 1, there exists an algorithm that uses the upper bound of $M$ and $k$ and ensures exact recovery of linear subspaces and hence the function $c_j$ with high probability. That is, for any large $n$, define*

$$\mathcal{E} := \Big\{\forall i \in [M], \exists j_i \in [M] \text{ s.t. } V_{j_i} = \widehat{V}_i\Big\}$$

*Here $\widehat{V}_i$ denotes the $i$-th estimated subspace of this algorithm $\mathsf{Alg}(n_0)$. Then*

$$\mathbb{P}[\mathcal{E}^c] \lesssim Mn^{-10}$$

*with the randomness taken over samples $\{X^{(i)}\}_{i=1}^{n_0}$, and here we take $n_0 = O(c_p^2 M^2(k+1) \log n)$.*

We will further implement an algorithm that proves Lemma 3 and discuss its complexity in Appendix B.1. This lemma is just for theoretical guarantee for our further score estimation under exact subspaces recovery.

With exact subspace recovery, when constructing the score component in (12), we only need to estimate $s_t^{\mathsf{low}}$ and the following lemma gives a time-dependent rate that only relies on intrinsic dimension $k_i$, using certain low-dimensional estimator (17).

**Lemma 4** (Low-dim score estimator). *For any $\sigma$ sub-gaussian distribution $\nu$ in $\mathbb{R}^k$, denote $p_t$ as the density of $\nu \star \mathcal{N}(0, tI_k)$ and $s_t = \nabla \log p_t$ as its score. Suppose that $\{X^{(i)}\}_{i=1}^N$ are $N$ i.i.d samples from $\nu$. Then for any $t > 0$, we could construct a kernel-based score estimator $\widehat{s}_t$ using (17) that satisfies,*

1. ***Time-dependent $L^2$ estimation error.***

$$\int_{\mathbb{R}^k} \mathbb{E}\left[\left\|\widehat{s}_t(x) - s_t(x)\right\|_2^2\right] p_t(x) \, \mathrm{d}x \lesssim \left(\frac{4}{\sqrt{\pi}}\right)^k \frac{1}{N} \left(\frac{1}{t} + \frac{\sigma^k}{t^{k/2+1}}\right) (\log N)^{k/2+2}$$

   *with expectation taken over samples $\{X^{(i)}\}_{i=1}^N$.*

2. ***Bounded estimator.***

$$\|\widehat{s}_t(x)\|_2 \leq \sqrt{\frac{2}{t} \log N}$$

Here we adopt the estimator from Cai & Li (2025) with a further cut-off. This achieves a similar rate w.r.t $t$ compared with Zhang et al. (2024); Cai & Li (2025). The proof is provided in Appendix B.3.

## B.1. Proof of Lemma 3.

Basically, subspace clustering for noise-free model aims to solve the following optimization problem iteratively,

$$\min_A \sum_{i=1}^{n_0} \|X^{(i)} - AA^\top X^{(i)}\|_0$$

Here the $L_0$ norm is defined as,

$$\|x\|_0 = \begin{cases} 0, & x = 0 \\ 1, & \text{else} \end{cases}$$

As a non-convex and non-smooth optimization problem, it is basically an NP-hard problem.

With known upper bound of both intrinsic dimension $k$ and the number of subspaces $M$, the following algorithm is guaranteed to recover exact subspace recovery with high probability and under Assumption 1,

Define the event sets,

$$\mathcal{E}_0 := \left\{\exists \text{ at least } k+1 \text{ samples on each subspace}\right\}$$

$$\mathcal{E}_1(p) := \left\{x \in \mathrm{span}\{y_1, \cdots, y_p\}, V_j \nsubseteq \mathrm{span}\{y_1, \cdots, y_p\}, \forall j \in [M]\right\}, \quad \text{for } p \leq \widetilde{k}$$

To ensure that Algorithm 2 works with probability larger than $1 - Mn^{-10}$, we only need,

$$\mathbb{P}[\mathcal{E}_0^c] \lesssim Mn^{-10} \tag{31a}$$

$$\mathbb{P}_{x, y_1, \cdots y_p \overset{\text{i.i.d}}{\sim} \mu^*}[\mathcal{E}_1(p)] = 0, \quad \forall p \leq \widetilde{k} \tag{31b}$$

Since under $\mathcal{E}_1(p)^c$, samples from other subspaces will be excluded via finding the smallest $p$, we could recover the exact subspace in each iteration.

---

**Algorithm 2** Exact subspace recovery

---

**Require:** Samples $\{X^{(i)}\}_{i=1}^{n_0}$, upper bound of the number of subspaces $\widetilde{M} \geq M$, upper bound of the intrinsic dimension $\widetilde{k} \geq k$

1: **for** j=1,..,$\widetilde{M}$ **do**

2:     Iterate over all separations of remained samples into 2 categories with one having $\widetilde{k} + 1$ samples until finding a case that these $\widetilde{k} + 1$ samples are linearly dependent.

3:     Find the smallest $p$, such as there exists $p + 1$ samples from these $\widetilde{k} + 1$ samples that are linearly dependent.

4:     Define $\widehat{V}_j$ as the span of these $p + 1$ points.

5:     Delete those samples that are on $\widehat{V}_j$.

6: **end for**

---

**Proof of (31b).**   We first conditioned on $y_1, \cdots, y_p$ and $V_j \not\subseteq \text{span}\{y_1, \cdots, y_p\}$ for all $j \in [M]$, then,

$$\mathbb{P}[X \in \text{span}\{y_1, \cdots, y_p\}] = \sum_{j=1}^{M} \int_{V_j \cap \text{span}\{y_1, \cdots, y_p\}} p_j^{\star}(\mathrm{d}x)$$

$$= 0 \quad (\text{Assumption 1})$$

Then we integrate this over $y_1, \cdots, y_p \sim \mu^*$ and get $\mathbb{P}[\mathcal{E}_1(p)] = 0$.

**Proof of (31a).**   Basically, under Assumption 1, denote $N_i(n_0), \forall i \in [M]$ as the random variable of the number of samples on $V_i$ with sample size $n_0$. Then, we could view $N_i(n_0)$ as the sum of Bernoulli r.v with $p = p_i^{\star}(V_i) \geq \frac{1}{c_\mu M}$ and thus,

$$N_i(n_0) \geq \frac{n_0}{2c_\mu M}, \quad \text{with probability } \geq 1 - 2e^{-\frac{n_0}{2c_\mu^2 M^2}}$$

here we apply Hoeffding inequality just like the proof of Claim 1. Finally, we could apply union bound and take,

$$n_0 = O(c_p^2 M^2 (k+1) \log n).$$

The iteration complexity for Algorithm 2 is bounded by $\binom{n_0}{\widetilde{k}+1} \cdot M$.

## B.2. Proof of Lemma 1.

The following lemma helps prove this result,

**Lemma 5** (MSE for $\widehat{p}_t(x)$ and $\widehat{q}_t(i,x)$)**.** *For kernel-based estimators $\widehat{p}_t(x)$ and $\widehat{q}_t(i,x)$ in (18a), (18b), it holds that,*

1. *They are unbiased estimators for $p_t(x)$ and $q_t(i,x)$ respectively under Assumption 1, i.e.,*

$$\mathbb{E}[\widehat{p}_t(x)] = p_t(x)$$
$$\mathbb{E}[\widehat{q}_t(i,x)] = q_t(i,x), \quad \forall i \in [M]$$

2. *Under Assumption 1, we have the following point-wise MSE bound,*

$$\mathbb{E}\left[\left(\widehat{p}_t(x) - p_t(x)\right)^2\right] \leq \frac{1}{(2\pi t)^{d/2} N} \sum_{i=1}^{M} e^{-\frac{1}{2t}\|x - \text{proj}_i(x)\|_2^2} \cdot q_t(i,x) \tag{33a}$$

$$\mathbb{E}\left[\left(\widehat{q}_t(i,x) - q_t(i,x)\right)^2\right] \leq \frac{1}{(2\pi t)^{d/2} N} e^{-\frac{1}{2t}\|x - \text{proj}_i(x)\|_2^2} q_t(i,x), \quad \forall i \in [M] \tag{33b}$$

We leave the proof of this lemma in the end of this section, and we first prove Lemma 1 as follows using Lemma 5. For $x \in \mathcal{G}_t(i)$

$$\mathbb{E}\big[\big(w_t(i,x) - \widehat{w}_t(i,x)\big)^2\big] = \mathbb{E}\Big[\big(\frac{q_t(i,x)}{p_t(x)} - \frac{\widehat{q}_t(i,x)}{\widehat{p}_t(x)}\big)^2\Big]$$

$$\lesssim \mathbb{E}\Big[\big(\frac{q_t(i,x) - \widehat{q}_t(i,x)}{p_t(x)}\big)^2\Big] + \mathbb{E}\Big[\big(\frac{\widehat{q}_t(i,x)}{\widehat{p}_t(x)}\big)^2 \cdot \frac{(\widehat{p}_t(x) - p_t(x))^2}{p_t^2(x)}\Big]$$

- For the first term,

$$\mathbb{E}\Big[\big(\frac{q_t(i,x) - \widehat{q}_t(i,x)}{p_t(x)}\big)^2\Big] \leq \frac{1}{p_t^2(x)} \cdot \frac{1}{N}(2\pi t)^{-d/2}e^{-\frac{1}{2t}\|x - \mathsf{proj}_i(x)\|_2^2}q_t(i,x). \quad \text{(Lemma 5)}$$

- For the second term,

$$\mathbb{E}\Big[\big(\frac{\widehat{q}_t(i,x)}{\widehat{p}_t(x)}\big)^2 \cdot \frac{(\widehat{p}_t(x) - p_t(x))^2}{p_t^2(x)}\Big] \leq \mathbb{E}\Big[\big(\frac{\widehat{p}_t(x) - p_t(x)}{p_t(x)}\big)^2\Big]$$

$$\leq \frac{1}{p_t^2(x)} \cdot \frac{1}{(2\pi t)^{d/2}N}\Big(\sum_{i=1}^{M} e^{-\frac{1}{2t}\|x - \mathsf{proj}_i(x)\|_2^2} \cdot q_t(i,x)\Big) \quad \text{(Lemma 5)}$$

Therefore, it holds that,

$$\mathbb{E}\big[\big(w_t(i,x) - \widehat{w}_t(i,x)\big)^2\big] \lesssim \frac{1}{p_t^2(x)} \cdot \frac{1}{(2\pi t)^{d/2}N}\Big(\sum_{i=1}^{M} e^{-\frac{1}{2t}\|x - \mathsf{proj}_i(x)\|_2^2} \cdot q_t(i,x)\Big).$$

Then we prove Lemma 5, and the proof idea is quite similar to the proof of Lemma 3 in Cai & Li (2025).

**Proof of Lemma 5.**

- First prove that both are unbiased estimators,

$$\mathbb{E}[\widehat{p}_t(x)] = \mathbb{E}[\frac{1}{N}\sum_{j=1}^{N}\varphi_t(x - X^{(j)};d)] = \mathbb{E}_{X\sim p^\star}[\varphi_t(x - X;d)] = p_t(x)$$

$$\mathbb{E}[\widehat{q}_t(i,x)] = \mathbb{E}_{X\sim p^\star}[\varphi_t(x - X;d)\mathbb{1}_{\{c(X)=i\}}] = \mathbb{E}_{X\sim p^\star}[\varphi_t(x - X;d)\mathbb{1}_{\{X\in V_i\}}] = \int_{V_i}\varphi_t(x - y;d)p_i^\star(\mathrm{d}y)$$

In the last equation, we apply Assumption 1.

- Then derive the mean squared error (MSE) for both estimators, i.e., both variance. To prove (33a), observe that,

$$\mathsf{Var}(\widehat{p}_t(x)) = \frac{1}{N}\mathsf{Var}_{X\sim p^\star}(\varphi_t(x - X;d)) \leq \frac{1}{N}\mathbb{E}_{X\sim p^\star}[\varphi_t^2(x - X;d)] = \frac{1}{N}\sum_{i=1}^{M}\int_{V_i}\varphi_t^2(x - y;d)p_i^\star(\mathrm{d}y).$$

Plug in the structure of $V_i$:

$$\int_{V_i}\varphi_t^2(x - y;d)p_i^\star(\mathrm{d}y) = \int_{V_i}(2\pi t)^{-d}e^{-\frac{1}{t}\|x - y\|_2^2}p_i^\star(\mathrm{d}y)$$

$$= (2\pi t)^{-d}e^{-\frac{1}{t}\|x - \mathsf{proj}_i(x)\|_2^2}\int_{V_i}e^{-\frac{1}{t}\|y - \mathsf{proj}_i(x)\|_2^2}p_i^\star(\mathrm{d}y)$$

Notice that,

$$\int_{V_i}e^{-\frac{1}{t}\|y - \mathsf{proj}_i(x)\|_2^2}p_i^\star(\mathrm{d}y) \leq \int_{V_i}e^{-\frac{1}{2t}\|y - \mathsf{proj}_i(x)\|_2^2}p_i^\star(\mathrm{d}y).$$

Hence,

$$\int_{V_i} \varphi_t^2(x-y;d)p_i^\star(\mathrm{d}y) \le (2\pi t)^{-d/2}e^{-\frac{1}{2t}\|x-\mathsf{proj}_i(x)\|_2^2} \cdot q_t(i,x).$$

Therefore,

$$\mathbb{E}\big[\big(p_t(x)-\widehat{p}_t(x)\big)^2\big] \le \frac{1}{N}\sum_{i=1}^{M}\int_{V_i}\varphi_t^2(x-y;d)p_i^\star(\mathrm{d}y)$$

$$\le \frac{1}{(2\pi t)^{d/2}N}\sum_{i=1}^{M}e^{-\frac{1}{2t}\|x-\mathsf{proj}_i(x)\|_2^2} \cdot q_t(i,x).$$

- To prove (33b), similarly,

$$\mathbb{E}\big[\big(q_t(i,x)-\widehat{q}_t(i,x)\big)^2\big] \le \frac{1}{N}\mathbb{E}_{X\sim p^\star}[\varphi_t^2(x-X;d)\mathbb{1}_{\{X\in V_i\}}] = \frac{1}{N}\sum_{j=1}^{M}\int_{V_j}\varphi_t^2(x-y;d)\mathbb{1}_{\{y\in V_i\}}p_j^\star(\mathrm{d}y)$$

Under Assumption 1, it holds that,

$$\mathbb{E}\big[\big(q_t(i,x)-\widehat{q}_t(i,x)\big)^2\big] \le \frac{1}{N}\int_{V_i}\varphi_t^2(x-y;d)p_i^\star(\mathrm{d}y) \le \frac{1}{N}(2\pi t)^{-d}e^{-\frac{1}{t}\|x-\mathsf{proj}_i(x)\|_2^2}\int_{V_i}e^{-\frac{1}{t}\|y-\mathsf{proj}_i(x)\|_2^2}p_i^\star(\mathrm{d}y)$$

$$\le \frac{1}{N}(2\pi t)^{-d}e^{-\frac{1}{t}\|x-\mathsf{proj}_i(x)\|_2^2}\int_{V_i}e^{-\frac{1}{2t}\|y-\mathsf{proj}_i(x)\|_2^2}p_i^\star(\mathrm{d}y)$$

$$\le \frac{1}{N}(2\pi t)^{-d/2}e^{-\frac{1}{2t}\|x-\mathsf{proj}_i(x)\|_2^2}q_t(i,x).$$

## B.3. Proof of Lemma 4.

In the proof of this lemma, the target distribution $\nu$ is a $\sigma$-subgaussian distribution in $\mathbb{R}^k$, and we denote $p_t$ as the density of $\nu * \mathcal{N}(0,tI_k)$ and $s_t(\cdot) := \nabla\log p_t(x)$ as its score. $\{X^{(i)}\}_{i=1}^N \overset{\text{i.i.d}}{\sim} \nu$ are N samples for score estimation.

We take the estimator based on (17) as,

$$\widehat{s}_t(x) = \mathsf{clip}_R\Big(\frac{\nabla\widehat{g}_t(x)}{\widehat{g}_t(x)}\psi\Big(\widehat{g}_t(x);\frac{\log N}{N(2\pi t)^{k/2}}\Big)\Big) \quad \text{with } R := \sqrt{\frac{2}{t}\log N}. \tag{34}$$

Here the clip operator is defined as,

$$\mathsf{clip}_r(z) := \begin{cases} z, & \|z\|_2 \le r \\ \mathsf{proj}_{B_r(0)}(z), & \text{else} \end{cases}$$

where $\mathsf{proj}_{B_r(0)}(z) = r \cdot \frac{z}{\|z\|_2}$ means the projection of $z$ on that ball; $\psi(x;\eta) := \mathbb{1}_{\{x\ge\eta\}}$ is the hard thresholding function; $\widehat{g}_t(x) := \frac{1}{N}\sum_{i=1}^N \varphi_t(X^{(i)}-x;k)$ is the kernel based density estimator. As a remark, the estimator before clipping is almost the same as that in Cai & Li (2025), and we apply hard-thresholding here for simplicity since we use DDPM sampling procedure.

Then we show that the score estimator defined in (34) satisfies the conditions in Lemma 4.

In Cai & Li (2025), they defined

$$\mathcal{F}_t := \Big\{x : p_t(x) \ge \frac{c_\eta\log N}{N(2\pi t)^{k/2}}\Big\}$$

and it holds that,

$$\int_{\mathcal{F}_t^c} p_t(x)\,\mathrm{d}x \lesssim \Big(\frac{16}{\pi}\Big)^{k/2}\frac{1}{N}\Big(1+\frac{\sigma^k}{t^{k/2}}\Big)(\log N)^{k/2+1} \tag{35a}$$

$$\int_{\mathcal{F}_t^c} \|s_t(x)\|_2^2 \cdot p_t(x)\,\mathrm{d}x \lesssim \Big(\frac{16}{\pi}\Big)^{k/2}\frac{1}{N}\Big(\frac{1}{t}+\frac{\sigma^k}{t^{k/2+1}}\Big)(\log N)^{k/2+2} \tag{35b}$$

from Lemma 8 in Cai & Li (2025). Notice that from Lemma 4 in Cai & Li (2025), it has been proven that,

$$\|s_t(x)\|_2^2 \le \frac{2}{t} \log \frac{1}{(2\pi t)^{k/2} p_t(x)}$$

Hence, for $x \in \mathcal{F}_t$, one has,

$$\|s_t(x)\|_2^2 \le \frac{2}{t} \log \frac{N}{c_\eta \log N} \le \frac{2}{t} \log N. \tag{36}$$

Here we use the constant selection of $c_\eta \ge 2$ as in Cai & Li (2025).

That's the reason why we use the clip operator in (34). Therefore, with the estimator constructed in (34), it automatically satisfies the bounded condition and for its $L^2$ error,

$$\int_{\mathbb{R}^k} \mathbb{E}\big[\big\|\widehat{s}_t(x) - s_t(x)\big\|_2^2\big] p_t(x)\,\mathrm{d}x \le \int_{\mathcal{F}_t} \mathbb{E}\big[\big\|\widehat{s}_t(x) - s_t(x)\big\|_2^2\big] p_t(x)\,\mathrm{d}x$$

$$+ \int_{\mathcal{F}_t^c} \mathbb{E}\big[\big\|\widehat{s}_t(x) - s_t(x)\big\|_2^2\big] p_t(x)\,\mathrm{d}x$$

$$=: \chi_1 + \chi_2$$

- **For $\chi_1$.** Due to (36), for $x \in \mathcal{F}_t$, $s_t(x) = \mathsf{clip}_R(s_t(x))$. What's more, $\mathsf{clip}_R(\cdot)$ is 1-Lip continous, and thus,

$$\chi_1 \le \int_{\mathcal{F}_t} \mathbb{E}\big[\big\| \frac{\nabla \widehat{g}_t(x)}{\widehat{g}_t(x)} \psi\big(\widehat{g}_t(x); \frac{\log N}{N(2\pi t)^{k/2}}\big) - s_t(x)\big\|_2^2\big] \cdot p_t(x)\,\mathrm{d}x$$

$$\le \int_{\mathbb{R}^k} \mathbb{E}\big[\big\| \frac{\nabla \widehat{g}_t(x)}{\widehat{g}_t(x)} \psi\big(\widehat{g}_t(x); \frac{\log N}{N(2\pi t)^{k/2}}\big) - s_t(x)\big\|_2^2\big] \cdot p_t(x)\,\mathrm{d}x$$

$$\le \frac{(4/\sqrt{\pi})^k}{N}\Big(\frac{1}{t} + \frac{\sigma^k}{t^{k/2+1}}\Big)(\log N)^{k/2+1}$$

  The last inequality results from Proposition 1 in Cai & Li (2025), which holds for hard thresholding $\psi$ by simple modification.

- **For $\chi_2$.** Notice that,

$$\chi_2 := \int_{\mathcal{F}_t^c} \mathbb{E}\big[\big\|\widehat{s}_t(x) - s_t(x)\big\|_2^2\big] p_t(x)\,\mathrm{d}x$$

$$\le \int_{\mathcal{F}_t^c} \Big(R^2 + \|s_t(x)\|_2^2\Big) p_t(x)\,\mathrm{d}x$$

$$\le R^2 \int_{\mathcal{F}_t^c} p_t(x)\,\mathrm{d}x + \int_{\mathcal{F}_t^c} \|s_t(x)\|_2^2 \cdot p_t(x)\,\mathrm{d}x$$

$$\lesssim \Big(\frac{16}{\pi}\Big)^{k/2} \frac{1}{N}\Big(\frac{1}{t} + \frac{\sigma^k}{t^{k/2+1}}\Big)(\log N)^{k/2+2} \quad (\text{Using } (35a) + (35b))$$

Therefore, it holds that,

$$\int_{\mathbb{R}^k} \mathbb{E}\big[\big\|\widehat{s}_t(x) - s_t(x)\big\|_2^2\big] p_t(x)\,\mathrm{d}x \lesssim \Big(\frac{4}{\sqrt{\pi}}\Big)^k \frac{1}{N}\Big(\frac{1}{t} + \frac{\sigma^k}{t^{k/2+1}}\Big)(\log N)^{k/2+2}.$$

### B.4. Proof of Lemma 2

Observe that, conditioned on $N_i$, we could obtain $N_i$ i.i.d samples from normalized distribution $p_i^\star$ to estimate its score function. We only need to prove (24). Notice that,

$$q_t(i, x) = \int_{V_i} \varphi_t(x - y; d) p_i^\star(\mathrm{d}y) = (2\pi t)^{-d/2} e^{-\frac{1}{2t}\|x - \mathsf{proj}_i(x)\|_2^2} \int_{V_i} e^{-\frac{1}{2t}\|\mathsf{proj}_i(x) - y\|_2^2} p_i^\star(\mathrm{d}y)$$

$$= (2\pi t)^{-d/2} e^{-\frac{1}{2t}\|x - \mathsf{proj}_i(x)\|_2^2} p_i^\star(V_i) \int_{\mathbb{R}^k} e^{-\frac{1}{2t}\|z - A_i^\top x\|_2^2} p_i^{\mathsf{low}}(\mathrm{d}z)$$

$$= (2\pi t)^{-(d-k_i)/2} e^{-\frac{1}{2t}\|x - \mathsf{proj}_i(x)\|_2^2} p_i^\star(V_i) \cdot p_t^{\mathsf{low}}(i, A_i^\top x)$$

here $p_t^{\text{low}}(i, \cdot)$ denotes the density function of $p_i^{\text{low}} * \mathcal{N}(0, tI_{k_i})$ in $\mathbb{R}^{k_i}$. Hence,

$$
\int_{\mathbb{R}^d} \mathbb{E}\big[\big\|\widehat{s}_t(i, x) - s_t(i, x)\big\|_2^2 \big| N_i\big] q_t(i, x)\, \mathrm{d}x
$$
$$
\leq p_i^\star(V_i) \int_{\mathbb{R}^d} \mathbb{E}\big[\big\|\widehat{s}_t^{\text{low}}(i, A_i^\top x) - s_t^{\text{low}}(i, A_i^\top x)\big\|_2^2 \big| N_i\big] (2\pi t)^{-(d-k_i)/2} e^{-\frac{1}{2t}\|x - A_i A_i^\top x\|_2^2} \cdot p_t^{\text{low}}(i, A_i^\top x)\, \mathrm{d}x
$$

Apply the same linear transform as in (28),

$$
\int_{\mathbb{R}^d} \mathbb{E}\big[\big\|\widehat{s}_t(i, x) - s_t(i, x)\big\|_2^2 \big| N_i\big] q_t(i, x)\, \mathrm{d}x
$$
$$
\leq p_i^\star(V_i) \int_{\mathbb{R}^d} \mathbb{E}\big[\big\|\widehat{s}_t^{\text{low}}(i, z_{1:k_i}) - s_t^{\text{low}}(i, z_{1:k_i})\big\|_2^2 \big| N_i\big] (2\pi t)^{-(d-k_i)/2} e^{-\frac{1}{2t}\|z_{k_i+1:d}\|_2^2} \cdot p_t^{\text{low}}(i, z_{1:k_i})\, \mathrm{d}z
$$
$$
= p^\star(V_i) \int_{\mathbb{R}^{k_i}} \mathbb{E}\big[\big\|\widehat{s}_t^{\text{low}}(i, z_{1:k_i}) - s_t^{\text{low}}(i, z_{1:k_i})\big\|_2^2 \big| N_i\big] \cdot p_t^{\text{low}}(i, z_{1:k_i})\, \mathrm{d}z_{1:k_i} \quad \text{(Tonelli's Theorem)}
$$
$$
\lesssim p_i^\star(V_i) \cdot \frac{(4/\sqrt{\pi})^{k_i}}{N_i} \Big(\frac{1}{t} + \frac{\sigma^{k_i}}{t^{k_i/2+1}}\Big) \cdot \text{poly}\log N \quad \text{(Lemma 4)}
$$

Finally,

$$
\int_{\mathbb{R}^d} \mathbb{E}\big[\big\|\widehat{s}_t(i, x) - s_t(i, x)\big\|_2^2 \mathbb{1}_{\{N_i \geq n_i\}}\big] q_t(i, x)\, \mathrm{d}x = \int_{\mathbb{R}^d} \mathbb{E}\Big[\mathbb{E}\big[\big\|\widehat{s}_t(i, x) - s_t(i, x)\big\|_2^2 \big| N_i\big] \cdot \mathbb{1}_{\{N_i \geq n_i\}}\Big] q_t(i, x)\, \mathrm{d}x
$$
$$
= \mathbb{E}\Big[\Big(\int_{\mathbb{R}^d} \mathbb{E}\big[\big\|\widehat{s}_t(i, x) - s_t(i, x)\big\|_2^2 \big| N_i\big] \cdot q_t(i, x)\, \mathrm{d}x\Big) \cdot \mathbb{1}_{\{N_i \geq n_i\}}\Big]
$$
$$
\lesssim p_i^\star(V_i) \cdot \frac{(4/\sqrt{\pi})^{k_i}}{n_i} \Big(\frac{1}{t} + \frac{\sigma^{k_i}}{t^{k_i/2+1}}\Big) \cdot \text{poly}\log N.
$$

## C. Proof of Claims

### C.1. Proof of Claim 1

For any $i \in [M]$, we first bound the probability $\mathbb{P}[N_i < \frac{N}{2c_p M}]$. Note that $N_i = \sum_{j=1}^{N} \mathbb{1}_{\{X^{(j)} \in V_i\}}$ is basically the sum of $N$ i.i.d Bernoulli r.vs with parameter $p^\star(V_i) \geq \frac{1}{c_p M}$ (Assumption 1). Applying Hoeffding's inequality, we have,

$$
\mathbb{P}\Big[N_i < \frac{N}{2c_p M}\Big] = \mathbb{P}\Big[N_i - N \cdot p^\star(V_i) < -\Big(N \cdot p^\star(V_i) - \frac{N}{2c_p M}\Big)\Big]
$$
$$
\leq \exp\Big(-2N \cdot \big(\frac{1}{2c_p M}\big)^2\Big) = \exp\Big(-\frac{N}{2c_p^2 M^2}\Big).
$$

Then applying union bound over $i \in [M]$, it holds that,

$$
\mathbb{P}\big[\mathcal{A}^c\big] = \mathbb{P}\Big[\exists i \in [M], N_i < \frac{N}{2c_p M}\Big] \leq M e^{-\frac{N}{2c_p^2 M^2}}.
$$

## C.2. Proof of Claim 2

For the target distribution $p^\star$, it is supported on $\cup_{i=1}^M V_i$. Hence for any $\theta \in \mathbb{R}^d$ and $\|\theta\|_2 = 1$, denote r.v $X \sim p^\star$

$$
\begin{aligned}
\mathbb{E}\Big[\exp\Big(\frac{(X^\top \theta)^2}{\sigma^2}\Big)\Big] &= \sum_{i=1}^M \int_{V_i} \exp\Big(\frac{(x^\top \theta)^2}{\sigma^2}\Big) p_i^\star(\mathrm{d}x) \\
&= \sum_{i=1}^M p_i^\star(V_i) \int_{\mathbb{R}^{k_i}} \exp\Big(\frac{(z^\top A_i^\top \theta)^2}{\sigma^2}\Big) p_i^{\mathsf{low}}(\mathrm{d}z) \\
&\leq \sum_{i=1}^M p_i^\star(V_i) \int_{\mathbb{R}^{k_i}} \exp\Big(\frac{(z^\top A_i^\top \theta)^2}{\sigma_i^2}\Big) p_i^{\mathsf{low}}(\mathrm{d}z) \\
&\leq \sum_{i=1}^M p_i^\star(V_i) \cdot 2 = 2.
\end{aligned}
$$

Here we use $\|A_i^\top \theta\| \leq 1$ and Assumption 2. This shows that, $p^\star$ is a sub-gaussian distribution in $\mathbb{R}^d$ with parameter $\sigma$. Therefore, it is straightforward that $p^\star * \mathcal{N}(0, tI_d)$ is $c\sqrt{\sigma^2 + t}$ sub-gaussian, for some absolute constant $c > 0$.

- For $\int_{\mathcal{B}_t^c} p_t(x)\mathrm{d}x$, it holds that,

$$
\begin{aligned}
\int_{\mathcal{B}_t^c} p_t(x)\mathrm{d}x &\leq \sum_{i=1}^M \mathbb{P}_{Z_t \sim p_t}\big[\|A_i^\top Z_t\|_2 > B_t\big] \\
&\leq \sum_{i=1}^M \mathbb{P}_{Z_t \sim p_t}\big[\|A_i^\top Z_t\|_\infty > B_t/\sqrt{k_i}\big] \lesssim Mk \exp\Big(\frac{cB_t^2}{k_i(\sigma^2 + t)}\Big) \lesssim \frac{Mk}{N^4} \quad \text{(Lemma 6)}
\end{aligned}
$$

Recall the definition of $\mathcal{B}_t$ in (27), and we take large enough constant $C_B > 0$ to ensure $\exp\big(\frac{cB_t^2}{k_i(\sigma^2 + t)}\big) \lesssim N^{-4}$.

- For $\int_{\mathcal{B}_t^c} \|x\|_2^2 \cdot p_t(x)\mathrm{d}x$, it holds that,

$$
\begin{aligned}
\int_{\mathcal{B}_t^c} \|x\|_2^2 \cdot p_t(x)\mathrm{d}x &= \mathbb{E}_{Z_t \sim p_t}[\|Z_t\|_2^2 \mathbb{1}_{\{Z_t \notin \mathcal{B}_t\}}] \\
&\leq \sqrt{\mathbb{E}_{Z_t \sim p_t}[\|Z_t\|_2^4] \cdot \int_{\mathcal{B}_t^c} p_t(x)\mathrm{d}x} \quad \text{(C-S Ineq)} \\
&\lesssim \sqrt{d^2(\sigma^2 + t)^2 \frac{Mk}{N^4}} \\
&\lesssim \frac{d\sqrt{M}}{N^2}(\sigma^2 + t).
\end{aligned}
$$

# D. Auxiliary Lemmas

**Lemma 6** (Tail bound for subgaussian random vectors). *Let $\nu$ be a $\sigma$-subgaussian distribution in $\mathbb{R}$, i.e,*

$$
\sigma = \|X\|_{\psi_2} := \inf\Big\{t > 0 : \mathbb{E}\exp\big(X^2/t^2\big) \leq 2\Big\}, \quad X \sim \nu. \tag{37}
$$

*Then we have the following tail bound for $X \sim \nu$, any $r \geq 0$ and some absolute constant $c > 0$,*

$$
\mathbb{P}[|X| \geq r] \leq 2\exp\big(-\frac{cr^2}{\sigma^2}\big) \tag{38}
$$

$$
\mathbb{E}[|X|^2 \mathbb{1}_{\{|X| \geq r\}}] \leq 2(r^2 + \sigma^2/c)\exp\big(-\frac{cr^2}{\sigma^2}\big). \tag{39}
$$

*Proof.* • **Proof of (38).** This is a classical result for sub-gaussian r.v. We can find the proof in Proposition 2.5.2 in (Vershynin, 2020).

• **Proof of (39).** For any $r \geq 0$,

$$
\begin{aligned}
\mathbb{E}\big[|X|^2 \mathbb{1}_{\{|X| \geq r\}}\big] &= \int_{\mathbb{R}} x^2 \mathbb{1}_{\{|x| \geq r\}} \cdot p(x) \mathrm{d}x \\
&= \int_{\mathbb{R}} \Big( \int_0^{|x|} 2z dz \Big) \mathbb{1}_{\{|x| \geq r\}} p(x) \mathrm{d}x \\
&= 2 \int_{\mathbb{R}} \int_0^{+\infty} z \cdot \mathbb{1}_{\{|x| \geq z\}} \mathbb{1}_{\{|x| \geq r\}} p(x) \mathrm{d}z \mathrm{d}x \quad \text{(Tonelli's Theorem)} \\
&= 2 \int_0^{+\infty} \Big( \int_{\mathbb{R}} \mathbb{1}_{\{|x| \geq r \vee z\}} \cdot p(x) \mathrm{d}x \Big) z \mathrm{d}z \quad \text{(Tonelli's Theorem)} \\
&\leq r^2 \mathbb{P}[|X| \geq r] + 2 \int_r^{+\infty} z \cdot \mathbb{P}[|X| \geq z] \mathrm{d}z \\
&\leq r^2 \mathbb{P}[|X| \geq r] + 4 \int_r^{\infty} z \cdot e^{-cz^2/\sigma^2} \mathrm{d}z \\
&= 2(r^2 + \sigma^2/c) \exp\Big( -\frac{cr^2}{\sigma^2} \Big).
\end{aligned}
$$

$\square$

