# OpenReview forum: "Diffusion Models Are Statistically Optimal for Learning Low-Dimensional Multi-Modal Distributions"
_ICML.cc/2026/Conference — ICML 2026 regular_

### Official Review · Reviewer_9BJj · 2026-03-05

**Soundness:** 2
**Presentation:** 1
**Significance:** 1
**Originality:** 1
**Overall Recommendation:** 4
**Confidence:** 3

**Summary:**

The paper studies end-to-end guarantees for diffusion models when the target distribution is supported on a union of low-dimensional linear subspaces. The authors establish near-optimal minimax sample complexity bounds for achieving $\varepsilon$ error in 1-Wasserstein distance.

The proposed estimator consists of three main steps: (i) recovering the underlying subspaces via clustering methods, (ii) estimating the score of the Gaussian-smoothed distribution within each subspace using kernel density estimation, and (iii)estimating the mixture weights of the different components. These estimates are then combined to construct a score estimator used in a diffusion sampling procedure.

**Compliance With Llm Reviewing Policy:**

Affirmed.

**Final Justification:**

The rebuttal addresses most of the points raised in my review. In particular, the clarification regarding the minimax optimality claim is convincing: the apparent gap was due to a typo rather than a logical error. The positioning relative to related work has also been substantially improved. I find the sketch of the extension to distributions approximately supported on a union of low-dimensional subspaces to be promising, though the quantitative conditions under which the minimax rate is preserved remain to be made precise. On this basis, I raise my score to weak acceptance.

**Key Questions For Authors:**

- **More precise positioning of related work**
  Many related works are discussed in detail; however, I would like to ask the authors to connect more clearly what is stated in the introduction with the discussion in the related works section. For instance, in the introduction it is stated:

  >While these works demonstrate improved sample complexities that depend on some intrinsic dimension, their results require strong assumptions on the data distribution. A prominent example is requiring the density to be uniformly bounded away from zero on its support.

  This phrasing may give the impression that the present paper is the first to avoid such assumptions. However, related work such as *Chen et al. (2023)* appears to establish results without requiring a lower bound on the density. Likely only some of the works cited have that lower-bound assumption (such as *Azangulov et al. (2025)*). The authors should clarify more precisely how their assumptions differ from each prior work. I am especially interested in *Cai & Li (2025)*, *Chen et al. (2023)*, and *Atanasov et al. (2024)*.

  I remark that these references are already discussed quite well in the paper; I would just like more details to better assess the significance of the present contribution, since navigating all the technical assumptions can be error-prone. My understanding is that each of these works has slightly different assumptions and recovers slightly different rates. Could the authors clarify exactly how the present submission generalizes each of those works?

- Connected to the fact that the data distribution strictly lies on a union of low-dimensional subspaces: the score estimation still works once the data is convolved with a $d$-dimensional Gaussian, becoming full-dimensional. Does this mean that the argument could be adapted to initial densities that are only close to the subspaces (i.e., similar to how $p_t$ looks for small positive times)? I imagine there could be difficulties in estimating the subspaces starting from samples of $p_t$, but for very small times it should work. Can the authors give an idea of how much noising can be tolerated before the techniques break down?

**Limitations:**

There is no focused discussion of the limitations. However, this is a theoretical work, so most of the limitations can be found in the assumptions and the in the statements of the theorems.

**Strengths And Weaknesses:**

## Strengths

- **Addresses an important theoretical question.**
  Providing end-to-end guarantees for diffusion models is a challenging problem. Establishing statistical benchmarks in terms of sample complexity is valuable for improving our understanding of the theoretical foundations of modern generative modeling and self-supervised learning.

- **Nearly optimal minimax guarantees.**
  The paper derives nearly sharp minimax rates for the sample complexity required to achieve a given error in 1-Wasserstein distance. Such optimality results are valuable for clarifying the statistical limits of diffusion-based generative models.

- **Relevance of rates with linear dependence on $d$**.
  Even though the data lies on low dimensional subspaces, the diffusion happens in the high dimensional space $\mathbb R^d$, hence the analysis cannot be completely reduced to a genuinely low dimensional problem. This point highlights the relevance of recovering sample complexity bounds that do not depend exponentially on $d$.

- **Extension to structured data distributions.**
  The analysis considers data supported on a union of low-dimensional subspaces, which introduces nontrivial geometric structure beyond the settings typically studied in previous analyses. The work also avoids assumptions such as requiring the density to be uniformly bounded away from zero on its support.

- **Conceptually clear estimation pipeline.**
  The proposed procedure decomposes the problem into subspace recovery, component-wise score estimation, and mixture weight estimation. This modular structure makes the analysis relatively transparent and highlights how structural assumptions on the data can translate into improved statistical rates.


## Weaknesses

- **Restrictive data model.**
  The analysis assumes that the data distribution is supported exactly on a union of low-dimensional linear subspaces. While this allows the authors to obtain nearly sharp minimax rates, it is a rather limited model for real data. In many practical settings, data is only approximately supported on a low-dimensional structure, with additional ambient noise. In such cases, identifying the subspaces becomes significantly more challenging. It would be more compelling if the analysis could accommodate noise around the subspaces or approximate low-dimensional structure.



- **Limited technical novelty relative to existing frameworks.**
  The overall proof strategy appears to follow the now standard template used in several recent works on the statistical analysis of diffusion models: score estimation via kernel density methods combined with an end-to-end analysis of the diffusion sampler. In particular, the arguments seem closely related to techniques used in works such as *(Cai & Li, 2025)*, *(Chen et al., 2023)*, and *(Oko et al., 2023)*. While the union-of-subspaces setting is an interesting extension, I believe that the technical innovations are limited, and the bulk of the work consists in the careful application of known tools.

- **Not complete engagement with closely related published results.**
  Two highly relevant papers deserve explicit discussion.
  - *Shallow Diffusion Networks Provably Learn Hidden Low-Dimensional Structure* (Boffi et al., ICLR 2025) shows that shallow neural diffusion networks can provably adapt to hidden low-dimensional subspace structure without density lower bounds and without architectural constraints. It would be helpful to clarify how the present results compare to this work, particularly in the special case \(M=1\).
  - *Adapting to Unknown Low-Dimensional Structures in Score-Based Diffusion Models* (Li & Yan, NeurIPS 2024) establishes intrinsic-dimension-dependent guarantees for DDPM sampling under unknown low-dimensional structure. The relationship between Theorem 2 in the present submission and the guarantees in that work should be explicitly discussed.


- **Numerical results are very limited.**
  The numerical simulations in Section 5 are very limited. In particular, there are no error bars indicating the variance of the points shown in Figure 1. As presented, the experiments add very little to the work, whereas they could have been an important section in which realistic (or at least more complex) diffusion models are compared with the statistical baselines. Without such a comparison there is essentially no connection to practice, which seems like a missed opportunity since it emerges for cited works that ERM analysis performed on neural network architectures could reach good rates, close to minimax.

- **Limited diffusion insight.**
  The main focus of the work is score estimation; the diffusion component appears mainly as a wrapper around it. This should be stated more clearly and slightly impacts the perceived significance of the contribution.

- **Ad-hoc estimator**.
   The authors remark that
>the kernel-based score estimator developed in this paper is primarily a theoretical proof device.

   and
   >we do not position this estimator as a practical alternative to neural network-based approaches.

   This is fair for a theoretical work that aims to compute statistical baselines. However, it weights on the significance that the algorithm proprosed is constructed with prior knowledge of the data generation process. Clustering is of course a common pre-processing step, but then the use of Gaussian kernels  and the splitting in "score component estimator" and "weight estimator" are clearly specifically designed for the task.


- **Claim of minimax optimality requires clarification.**
  The paper argues that the proposed sampling algorithm is near-minimax optimal because the minimax risk of estimating a \(k\)-dimensional density scales as \(n^{-1/(k \vee 2)}\) (Chewi et al., 2024), and density estimation is harder than sampling. However, this reasoning does not directly establish optimality for the sampling problem. The result of Chewi et al. provides a lower bound for density estimation, which only implies that the minimax risk for sampling is at most that rate. In principle, the minimax sampling rate could be strictly faster. Therefore, the current argument does not rule out the possibility that sampling can be achieved at a faster rate than density estimation. The authors should clarify this point and explain whether any lower bound specific to the sampling problem is known.

- Figure 1 is very small and difficult to read. Increasing the size of the figure and the font of the labels and legend would improve readability.

- After Equation (1), the text refers to "total variance distance". Presumably the intended term is *total variation distance*, which should be clarified.

- The constant $C_1$ in Theorem 1 does not appear in the equations, while $c_\mu$ appears instead. Is there a labeling inconsistency?

- Most of the time the rates are discussed only in terms of the factor $n^{-\frac{1}{k\vee 2}}$, but in Equation (20) of Theorem 2 a factor $d$ also appears. I suggest consistently stating the dependence on $d$ when discussing the rates.

- The bibliography appears to contain several duplicated entries, for example *(Tang & Yang, 2024a)* and *(Tang & Yang, 2024b)*, *(Chen et al., 2023b)* and *(Chen et al., 2023c)*, and *(Cai & Li, 2025a)* and *(Cai & Li, 2025b)*.


## Overall considerations

In its current state, the paper is not yet ready for publication due to the weaknesses detailed above. However, most of these issues could likely be addressed in the rebuttal, especially the discussion of missing related work, the clarification of the near-minimax optimality claim, and improvements to Figure 1 and the numerical section. If these points were adequately clarified, I believe the fundamental contributions of the paper could possibly reach the acceptance threshold.

---

> ### Author Rebuttal · Authors · 2026-03-30
>
> We thank the reviewer for the careful reading and the constructive feedback.
> ### Questions
> **Question 1: Positioning of related work.** To the best of our knowledge, **even in the single low-dimensional setting, our result is the first to achieve a (near-)optimal rate under only a subgaussian assumption on the target distribution, without extra assumptions on score regularity or density bounds.** Below we clarify the main differences from prior work.
> 1.  (Chen et al 2023) studies distributions supported exactly on a low-dimensional subspace and assumes the latent score of the forward process is Lipschitz for all time. This strong smoothness assumption significantly simplifies the analysis and excludes many practical distributions.
> 2.  (Azangulov et al 2024) assumes the target density is uniformly lower and upper bounded on a low-dimensional manifold. This also significantly simplifies the analysis, effectively requiring regularity of the score and mainly covering roughly uniform distributions.
> 3.  (Cai \& Li 2025) studies subgassuain distributions with smooth densities, but only considers the ambient-dimensional setting. The estimator and analysis do not exploit low-dimensional structure, and the result suffers from the curse of dimensionality: the rate grows exponentially with the ambient dimension.
>
> In addition to these differences in the single-structure setting, a main technical novelty of our work is handling the multi-subspace structure. **Beyond estimating the score on each low-dimensional subspace, we must also combine these component-wise estimators using data-driven weights.** This requires a substantially more involved estimator and error analysis, with the central challenge of keeping the dependence on the ambient dimension as mild as possible. This difficulty is specific to the multi-subspace setting and requires nontrivial extensions.
>
> **Question 2 and Weakness 1: data model.** Due to space constraints, please see our response to the first reviewer (Reviewer 39GA) on "Noisy data model".
> ### Weaknesses
> **Limited technical novelty.** Please see our response to Question 1.
>
> **Engagement with related results.** When $M=1$, our result is closely related to (Boffi et al 2024). It assumes the latent score of the forward process is Lipschitz for all time, whereas we only require the target distribution to be subgaussian, which covers a broader class of distributions. (Li \& Yan 2024) focuses on the sampling speed of DDPM. It shows that the sampling convergence rate depends on the intrinsic dimension but treats the score estimation error as a black box. Our result offers complementary contributions from the statistical perspective: we show that the score estimation error and final sampling error also adapt to the intrinsic dimension. We will discuss this explicitly in the revision.
>
> **Numerical results.** Thank you for the suggestion. We will add error bars in the revision. In addition, the purpose of the synthetic experiment is to validate Theorem 1 in a controlled setting. Evaluating the Wasserstein distance for larger $d,k$ is computationally prohibitive, and for real-world datasets these quantities are generally intractable, making this controlled verification infeasible. We will make this motivation and limitation more explicit in the revision.
>
> **Limited diffusion insight.** We agree the main contribution is on score estimation and will make this clearer in the revision. As score estimation is the central statistical bottleneck in diffusion models, we view our result as providing theoretical justification for their statistical efficiency on structured data.
>
> **Ad-hoc estimator.** We agree the constructed score estimator is tailored to the assumed geometry. That said, the value of the analysis goes beyond this specific kernel-based estimator: by giving an explicit low-complexity construction, it isolates the statistical mechanisms needed to exploit multi-modal low-dimensional structure and provides a first step toward understanding neural score estimators. Due to space constraints, please see our response to the first reviewer (Reviewer 39GA) on "Connection to neural score models."
>
> **Minimax optimality.** Thank you for catching this typo. We intended to say that sampling is at least harder than density estimation. Indeed, suppose a sampler outputs a distribution $\hat p$ with $TV(\hat p, p)\leq\epsilon$. By drawing arbitrarily many samples from it, we can estimate $\hat p$ to arbitrary accuracy, yielding a density estimator for $p$ with error at most $\epsilon$. Hence, any lower bound for density estimation also implies a lower bound for sampling.
>
> **Detailed corrections.** Thank you for catching the typos. In the revision, we will enlarge Figure 1, correct total variation distance, fix the notation in Theorem 1 $(n_0 \ge C_1 c_p^2 M^2 (k+1)\log n)$, clarify the rate refers to the convergence rate wrt sample size $n$ while prefactors may depend on $d$ and $M$, and remove duplicated references.

---

> > ### Author Rebuttal · Reviewer_9BJj · 2026-04-02
> >
> > The rebuttal addresses most of the points raised in my review. In particular, the clarification regarding the minimax optimality claim is convincing: the apparent gap was due to a typo rather than a logical error. The positioning relative to related work has also been substantially improved. I find the sketch of the extension to distributions approximately supported on a union of low-dimensional subspaces to be promising, though the quantitative conditions under which the minimax rate is preserved remain to be made precise. On this basis, I raise my score to weak acceptance.

---

> > > ### Author Response · Authors · 2026-04-02
> > >
> > > Thank you again for your careful review of our paper and for updating your rating.

---

### Official Review · Reviewer_48SD · 2026-03-13

**Soundness:** 3
**Presentation:** 2
**Significance:** 3
**Originality:** 4
**Overall Recommendation:** 4
**Confidence:** 4

**Summary:**

**Strengths.**

This article attempts to present a notable theme in diffusion model theory: can score-based samplers achieve minimax-optimal rates when the target distribution has low-dimensional, multi-modal structure? The authors intend to investigate a central concept — that the sample complexity of diffusion models should scale with intrinsic dimension rather than ambient dimension. They formalize this for targets supported on a union of linear subspaces (UoS) with subgaussian tails per component. The main result shows that a kernel-based score estimator, combined with reverse-time SDE sampling, achieves W1 error of order $n^{−1/(k∨2)}$ (up to logs and a $d$ prefactor), where $k$ is the maximal subspace dimension.

**Weaknesses.**

- The paper abstract says the rate "depends only on the intrinsic dimension k," but the proved bound appears to  $dM^{3/2}·n^{−1/(k∨2)} \cdot \textnormal{poly}\log n$. If this is case, the paper should state it clearly.

- The theory covers continuous-time sampling only, no discretization. Section 2 mentions Euler-Maruyama for practice, but Theorem 2 doesn't apply to it.

- The UoS assumption requires exact support on subspaces, needed for the noiseless subspace recovery step (Lemma 3). Real data is approximately, not exactly, low-dimensional. It would strengthen the paper to discuss whether the analysis degrades gracefully under $\varepsilon-$thickened subspaces.

- The experiment section is a single 2D example $(k=1, d=2, M=4)$. This doesn't probe the headline claim. A straightforward experiment with much higher $d$ and lower $k$ would directly show whether sample complexity scales with k or d.

- $\theta_\min$ is introduced in Assumption 1 but never surfaces in any theorem. Its role is presumably hidden in constants related to subspace recovery — making this explicit would be helpful.

**Compliance With Llm Reviewing Policy:**

Affirmed.

**Final Justification:**

The authors' clarifications on the $d$-dependence and $\theta_{min}$ were helpful, but the lack of higher-dimensional experiments supporting the headline claim (sample complexity scaling with k, not d) remains a central gap. The promised revision experiments are not yet in the paper. Therefore, I maintain my positive score.

**Key Questions For Authors:**

- Can you point to a lower bound that includes d? And also include the exact result in the cited monograph for lower bound in the paper.
- Please see the weakness section.

**Limitations:**

Yes

**Strengths And Weaknesses:**

**Strengths.** The paper makes a clean theoretical contribution to an important question. Prior work on diffusion models with low-dimensional structure either assumed a single manifold/subspace or required density lower bounds that exclude multi-modal targets — both significant limitations. This paper handles multiple subspaces with well-separated modes and non-vanishing inter-component gaps. The assumptions are mild: subgaussian tails subsume bounded support, and no smoothness or log-concavity is needed. The score decomposition in Eq. (11)–(12), separating the problem into low-dimensional component scores and posterior weights, seems likely to be useful beyond this specific setting. The proof architecture is well-organized: Lemma 1 controls weight estimation, Lemma 2 controls per-component score estimation, and both feed cleanly into Theorems 1 and 2.

---

> ### Author Rebuttal · Authors · 2026-03-30
>
> We appreciate the reviewer’s thoughtful comments and careful evaluation.
> ### Questions.
>
> **Dependence on $d$ in lower bound.**
>
> The lower bounds we are aware of do not depend on the ambient dimension $d$, and the lower bound cited in the paper is $n^{-\frac{1}{(k\vee 2)}}$ as well. Therefore, we believe the current upper bound is likely suboptimal in its prefactor dependence on $d$.
>
> In our analysis, the remaining $d$-dependence comes from the multi-modal setting: on a high-probability region, the $\ell_2$ norm of the score function is dominated by the normal component, i.e., $||s_t(i,x)||_2$ is of the order $\sqrt{d-k}\asymp \sqrt{d}$.
> When we combine the component-wise score estimators through data-driven weights, the error analysis requires controlling the estimation error of these weights, which leads to a dependence on $||s_t(i,x)||_2$ and therefore $d$. This is the main source of the ambient-dimension term.
> In particular, in the single-subspace case ($M=1$), our result is independent of $d$.
>
> We therefore believe that the current dependence on $d$ is likely a proof artifact and may be improved by a more careful analysis. Because our main focus in this paper is the convergence rate with respect to the sample size $n$, we have not attempted to optimize the dependence on $d$ in the present analysis. Clarifying whether this dependence is truly necessary or can be removed is an interesting direction for future work.
>
> ### Weaknesses.
> **Dependence on $d$ in upper bound.**
>
> The phrase “depends only on the intrinsic dimension $k$” was intended to refer to the convergence rate w.r.t. the sample size $n$: the exponent is governed by $k$, rather than the ambient dimension $d$. We will clarify this point in the revision.
>
> **Discretization error.**
>
> Thank you for pointing this out. Prior diffusion theory has shown that once the score estimation error is controlled, discretization error is not the dominant term in the final sampling error bound (Zhang et al 2024; Cai \& Li 2025). In other words, the **discretization error only affects how fast the sampling error converges to the minimax rate**.  As the scope of this paper is the statistical guarantee, we therefore focus on the main statistical bottleneck of the problem, namely the score estimation error.
>
> That said, we agree that incorporating the discretization error of practical samplers, such as Euler--Maruyama, would make the theory more complete. However, obtaining a sharp discretization bound in Wasserstein distance under mild distributional assumptions remains open.  Existing results typically rely on stronger regularity or structural assumptions, such as global Lipschitz regularity of the score [1] or log-concavity of the target distribution [2]. Hence, they are not compatible with the level of generality considered in our paper, where we only impose a subgaussian assumption on the target distribution.
>
> Extending our result to include such a sampling convergence guarantee would therefore require additional technical ideas beyond the scope of the current work. We will clarify this limitation more explicitly in the revision and highlight discretization in Wasserstein distance as an important direction for future work.
>
> **Noisy data model.**
>
> Thank you for this question. Due to space constraints, please see our response to the first reviewer (Reviewer 39GA) on "Noisy data model".
>
> **Experiments.**
>
> We fully agree with this point. However, evaluating the Wasserstein distance for large $d$ and $k$ is computationally prohibitive, which limits the scale of experiments we can include in this theory-focused paper. We will make this limitation clearer in the revision and will try to add higher-dimensional synthetic experiments to better test the predicted rate.
>
> **Role of $\theta_{\min}$.**
>
> Thank you for this suggestion. The parameter $\theta_{\min}$ is used only in the subspace recovery step, where it ensures sufficient separation between different subspaces, as in (Heckel \& Bölcskei 2013; Park et al 2014).
> As long as it is sufficiently large to guarantee successful recovery, the score estimation depends only on the recovered subspaces and no longer on $\theta_{\min}$. We will make this point explicit in the revision.
>
> **Additional References.**
>
> [1] Wang, X. and Wang, Z. (2024). Wasserstein bounds for generative diffusion models with Gaussian tail targets. arXiv preprint arXiv:2412.11251.
>
> [2] Gao, X., Nguyen, H. M., and Zhu, L. (2023). Wasserstein convergence guarantees for a general class of score-based generative models. arXiv preprint arXiv:2311.11003.

---

> > ### Author Rebuttal · Reviewer_48SD · 2026-04-03
> >
> > I thank the authors for the detailed response. The clarifications on the $d$-dependence as a proof artifact, the role of $\theta_{\min}$, and the separation between score estimation and discretization error are helpful.
> >
> > That said, the experimental concern remains my primary reservation. The headline claim of the paper is that sample complexity scales with $k$ rather than $d$, and this is precisely the kind of result that needs empirical support beyond a single 2D example. Even if exact Wasserstein computation is prohibitive at scale, the authors should be able to design synthetic experiments at moderately high $d$ with $k \ll d$ using scalable Wasserstein approximations, that capture the predicted scaling behavior.
> >
> > I note the authors' intent to add higher-dimensional experiments in revision, but view this as a central weakness, not a minor point. We therefore maintain our current score.

---

> > > ### Author Response · Authors · 2026-04-03
> > >
> > > Thank you again for your time and suggestion. In the revision, we will add synthetic experiments for the case $d \gg k $ using scalable Wasserstein approximations.

---

### Official Review · Reviewer_39GA · 2026-03-23

**Soundness:** 3
**Presentation:** 4
**Significance:** 4
**Originality:** 3
**Overall Recommendation:** 5
**Confidence:** 3

**Summary:**

This paper studies diffusion-based sampling for target distributions supported on a union of low-dimensional subspaces. Each component distribution is assumed to be subgaussian on its own subspace, which allows the model class to cover multi-modal targets with well-separated modes.
The paper constructs a regularized kernel-based estimator for the score of the Gaussian-smoothed target distribution, proves an $L^2$ score estimation bound whose rate depends on the maximal intrinsic dimension $k$ rather than the ambient dimension $d$, and then derives an end-to-end $W_1$ sampling guarantee of order $\widetilde{O}(n^{-1/(k\vee 2)})$.
The main technical step is to exploit the union-of-subspaces structure so that the smoothed score can be handled through component-wise low-dimensional score estimation together with estimation of mixture weights. In this way, the paper extends earlier intrinsic-dimension results beyond the single-subspace or single-manifold setting and obtains a near-minimax statistical guarantee for diffusion-based sampling on multi-modal low-dimensional data.

**Compliance With Llm Reviewing Policy:**

Affirmed.

**Final Justification:**

The authors approach the reviewer's concerns in their rebuttal.  I will rase Significance score from 3 to 4.

**Key Questions For Authors:**

**Can the analysis be extended to approximately structured data, where the target distribution is concentrated near a union of subspaces rather than exactly supported on it?**
A positive answer, even at a high level or as a partial result, would make the contribution much more relevant to realistic data and would strengthen my assessment of significance.

**Can the authors provide a more concrete roadmap for connecting the present theory to practical neural score models?**
The discussion section identifies this as an important open direction, but the paper currently remains at a fairly abstract level.  A more concrete answer would significantly strengthen the practical significance of the paper.

**Limitations:**

yes

**Strengths And Weaknesses:**

### Strengths

**Soundness.**
From the viewpoint of mathematical structure, the paper is careful and coherent. The estimator is explicit, the decomposition of the score is used in a systematic way. I did not find an obvious gap in the main proof strategy at the level of the theorem statements and proof sketch. In particular, the paper does more than state an abstract existence claim. It gives a concrete construction and analyzes it in a nontrivial geometric setting.

**Presentation.**
The paper is generally well organized. The problem setting, assumptions, estimator construction, and main guarantees are presented in a natural order. I also appreciated that the authors state explicitly that the proposed score estimator is mainly a theoretical proof device rather than a practical replacement for standard neural estimators. This helps the reader understand the scope of the results.

**Significance.**
Although I do not read the paper as a result about the practical expressive power of standard diffusion models, I do think it has value as a theoretical extension of the current literature on structured distributions. The generalization of a single low-dimensional structure to a multi-modal union-of-subspaces model is meaningful. It addresses a gap between existing intrinsic-dimension analyses and the kind of multi-modal structure that often appears in applications.

**Originality.**
The paper combines existing ingredients in a nontrivial way although the individual parts are not all entirely new (e.g. score estimation with kernel regularization, low-dimensional structure in diffusion theory, and reverse-time stability bounds).  The treatment of mixture weights and component scores in one unified estimator, followed by an end-to-end sampling guarantee, seems to be a genuine advance over prior single-structure results.

### Weaknesses
There is an inescapable point where the strong statements in the main theorems are obtained by designing an estimator that already uses the intrinsic low-dimensional structure. For that reason, the result does not by itself show that standard diffusion models or standard neural score learners automatically provide intrinsic-dimension adaptation although it is not fatal.

Also, the estimator is highly idealized and is not close to the estimators used in modern large-scale diffusion model. This is acknowledged by the authors, but it still limits the practical scope of the conclusions. The gap between the theorem and standard neural implementations remains substantial.

---

> ### Author Rebuttal · Authors · 2026-03-30
>
> We thank the reviewer for the thoughtful and constructive comments.
> ### Questions.
> **Noisy data model.**
>
> We fully agree with this point.
> The current manuscript focuses on the noiseless setting in order to isolate the essential ingredients of low-dimensional structure and multi-modality, without the additional technical distractions introduced by noise. In this setting, component-wise score estimation and mixture-weight estimation can be separated most transparently.
>
> That said, our framework can be straightforwardly extended to distributions concentrated near a union of low-dimensional subspaces. The main additional challenge is subspace estimation from noisy observations: once reliable subspace estimates are available, the treatment of multi-modality is similar. Thus, to illustrate the key difference, let us consider the single-subspace case $M=1$, which already captures the essence of the extension. Suppose data is generated from a subspace with additive noise in the normal direction:
> $$
> X_0 = A Z +  (I- A A^\top)\epsilon, \quad   \epsilon \sim \mathcal{N}(0, v^2 I_d).
> $$
> Below, we briefly discuss the case where the goal is to learn the low-dimensional component $AZ$. The alternative goal is to learn the full-dimensional $X_0$, which can be handled by a similar analysis argument.
>
> Note that the marginal of the forward process now becomes
> $$
> X_t = X_0 + \sqrt t W_t = A(Z + \sqrt t A^\top W_t) + (I- A A^\top)(\epsilon + \sqrt t W_t).
> $$
> Thus, the marginal of the forward process preserves the tangential and normal decomposition, and its **score admits an analogous form: the subspace component is governed by the low-dimensional score and the normal component remains Gaussian with an enlarged variance due to the added noise.** This suggests that we construct the score estimator as
> $$
> \hat{s}_t(x) = \hat{A} \hat{s}_t^{\mathrm{low}}(\hat{A}^\top x) - \frac{x-\hat{A} \hat{A}^\top x}{\hat{v}^2 + t},
> $$
> where $\hat{A}$ and $\hat{v}^2$ are estimates of the subspace and noise level obtained from samples.
>
> The overall proof strategy then parallels the noiseless case. The new step is to control the mismatch between the intrinsic variable
> $A^{\top}X_0$ and the projected variable $\hat{A}^\top X_0$ used to construct low dimensional score estimator. In this case, we expect a $L^2$-score error bound of the form:
> $$
>      \int_{t_{\min}}^{t_{\max}} E \big[||\hat{s}_t(X_t) - s_t^\star(X_t)||_2^2 \big] d t \lesssim \int _{t _{\min}}^{t _{\max}} \frac{1}{N t^{(k\vee 2)/2+1}} d t + \frac{\sigma^2 + v^2}{t _{\min}} E[|| AA^{\top} - \hat{A}\hat{A}^\top ||^2].
> $$
> So the same intrinsic-dimensional term remains, with an additional subspace-recovery term depending on noise level $v$. **When the noise level $v$ is relatively small, this extra term is negligible and the minimax rate still holds.**
>
> In the revision, we will clarify this point and discuss this extension in more detail.
>
> **Connection to neural score models.**
>
> While our estimator is kernel-based, we believe that the present result serves as an important step toward understanding statistical guarantees of neural network (NN) score estimators. In particular, it serves two purposes: (i) it establishes achievability of the fundamental statistical limit in the low-dimensional multi-modal setting, and (ii) **it identifies the structural ingredients that analysis for NN score estimators need to capture and constructs an explicit low-dimensional target for NN approximation.**
>
> Specifically, the NN-based approach naturally studies an ERM estimator over an NN class and bounds its score estimation error (e.g., Oko et al 2023, Azangulov et al 2024). Such an analysis typically involves two components: approximation error and generalization error. The main difficulty is the approximation step: constructing a NN approximation for the target score whose complexity depends on the intrinsic dimension $k$ rather than the ambient dimension $d$.
>
> Our decomposition in (11)-(12), together with the constructed kernel-based estimator, provide a concrete roadmap for this approximation step.
> The decomposition expresses the target score as a weighted combination of simpler low-dimensional components.
> Hence, by treating the subspace matrices $A_i$ as network parameters, the problem reduces to approximating these low-dimensional scores with associated weights. **In particular, our kernel-based estimator construction makes these approximation targets explicit, and can be used to show the existence of a neural score estimator with complexity depending on the intrinsic dimension $k$.** In this sense, the present kernel-based construction not only proves the achievability of minimax optimality, but also provides an explicit low-dimensional representation of the score. We view this as a useful proof framework and a roadmap for future analyses of NN estimators.
>
> In the revision, we will provide a detailed discussion of this point.

---

> > ### Author Rebuttal · Reviewer_39GA · 2026-04-04
> >
> > The rebuttal addresses two of my main concerns and makes me more positive about the paper.
> > The discussion of the noisy setting makes the extension to approximately structured data look plausible. The single-subspace sketch is helpful, and the proposed extra error term gives a clearer picture of what changes in the analysis.
> > Also, the response on neural score models clarifies the intended scope of the paper. I now better understand the current result as an achievability theorem for a structure-aware estimator, together with a roadmap for future analysis of neural score models. This resolves a significant part of my earlier concern about interpretation.

---

> > > ### Author Response · Authors · 2026-04-04
> > >
> > > Thank you again for your thoughtful review and constructive feedback!

---

### Decision · Program_Chairs · 2026-04-30

**Decision:**

Accept (regular)

**Comment:**

The paper studies diffusion-based sampling for target distributions supported on a union of low-dimensional subspaces, where each component distribution is assumed to be subgaussian on its own subspace. The authors propose a kernel-based estimator for the score of the Gaussian-smoothed target distribution, prove a score estimation bound whose convergence rate depends on the intrinsic dimension $k$ rather than the ambient dimension $d$, and derive an end-to-end sampling guarantee in the $1$-Wasserstein distance.

Reviewers are generally positive about the theoretical analysis for diffusion models on multi-modal data presented in the paper. Reviewer
39GA notes that the score estimator being studied is kernel-based and not the neural network (NN) estimators used in practice (as acknowledged by the authors themselves). The authors claim that their current approach would provide a roadmap for the analysis of NN estimators, though this obviously would require another work. Reviewer 48SD notes that the ambient dimension $d$ still appears in the bound, though not part of the convergence rate. The authors argue that this is likely an artifact of the proof and could be improved. Reviewer 48SD  and 9BJj are concerned that the experimental section is very limited. There is only one $2D$ experiment, with $d=2$, $k=1$. As suggested by Reviewer 48SD, an experiment with a much higher $d$ and lower $k$ would show much better whether the sample complexity scales with $d$ or with $k$.

The scores are Weak Accept, Weak Accept, Accept.